# Toward Practical Learning-based Frequency Estimation without Ground Truth

## Abstract

Estimating the frequency of items on the high-volume, fast data stream has been extensively studied in many areas, such as database and network measurement. Traditional sketch algorithms only allow to give very rough estimates with limited memory cost, whereas some learning-augmented algorithms have been proposed recently, their offline framework requires actual frequencies that are challenging to access in general for training, and speed is too slow for real-time processing, despite the still coarse-grained accuracy. To this end, we propose a more practical learning-based estimation framework namely UCL-sketch, by following the line of equation-based sketch to estimate per-key frequencies. In a nutshell, there are two key techniques: online training via equivalent learning without ground truth, and highly scalable architecture with logical estimation buckets. We implemented experiments on both real-world and synthetic datasets. The results demonstrate that our method greatly outperforms existing state-of-the-art sketches regarding per-key accuracy and distribution, while preserving resource efficiency. Our code is attached in the supplementary material, and will be made publicly available.

## 1 Introduction

The frequency or volume estimation of unending data streams is a concern in many domains, starting with telecommunications but spreading to social networks, finance, and website engine. In network fields, for example, professionals want to keep track of the activity frequency to identify overall network health and potential anomalies or changes in behavior, which, however, is often challenging because the amount of information may be too large to store in an embedded device or to keep conveniently in fast storage (Cormode, 2017). As a consequence, *sketch*, which is a set of counters or bitmaps associated with hash functions, and a set of simple operations that record approximate information (Yang et al., 2018b), has grown in popularity in the context of high-velocity data streams and limited computational resources. Such an approximate algorithm is much faster and more efficient, yet this comes at the expense of unsatisfactory accuracy and cover proportion, especially when facing unbalanced stream characteristics, such as a Zipf or Power-law distribution (Kumar et al., 2004; Roy et al., 2016).

On the other hand, recent years have witnessed the integration of deep learning technology with numerous classic algorithms: index (Kraska et al., 2018), bloom filter (Mitzenmacher, 2018; Rae et al., 2019), caching (Lykouris & Vassilvitskii, 2021), graph optimization (Khalil et al., 2017; Feng et al., 2023) and so on. In particular, the research about learning-augmented streaming algorithms (Hsu et al., 2019; Jiang et al., 2020; Du et al., 2021; Yan et al., 2022; Cheng et al., 2023; Aamand et al., 2024) is receiving significant attention due to the powerful potential of machine learning (ML) to relieve or eliminate the binding of data characteristics and the sketch design. Their typical workflow involves training a heavy hitter oracle, which receives a key and returns a prediction of whether it will be heavy or not, then inserts the most frequent keys into unique buckets and applies a sketch to the remaining keys. Although filtering heavy items has been proven to improve the overall sketch performance on heavy-tailed distribution (Roy et al., 2016; Hsu et al., 2019), these offline and supervised methods could hardly work in real-world applications. First, an unavoidable difficulty in designing algorithms in the learned sketch model is that ground truth like actual frequencies or labels for which key is large are not known in advance (Jiang et al., 2020). Moreover, since their models are only fitted on the past data, the prediction performance of the oracle tends to deteriorate rapidly over time. That is to say, the neural network must be retrained with new labeled datasets frequently,

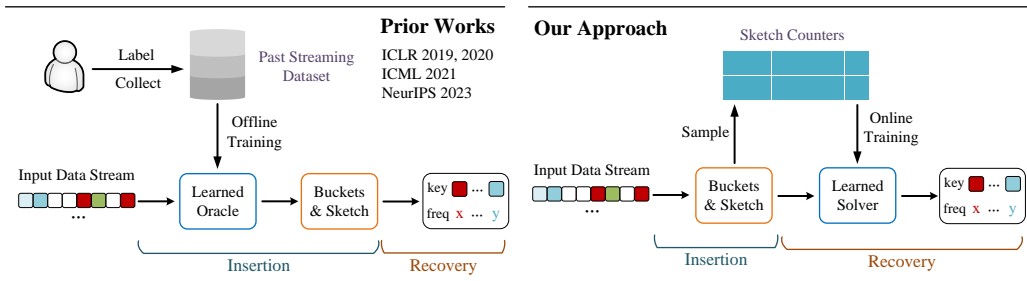

Figure 1: **Comparison between the previous learning-augmented sketches and our studied learning-based sketch:** In our approach, we empower the sketch with learning technologies in the recovery phase to improve streaming throughput. The model is online trained using just compressed counters in the sketch, which is much more practical and efficient than the prior works.

therefore, all the above-mentioned sketches face a common problem in terms of updating the out-of-date classifier (Li et al., 2020). Besides, the idea of passing a deep model to reduce conflicts also incurs more insertion time and space cost, which may not be advisable for hashing-intense situations by considering data stream is processed sequentially in only one pass.

Building upon the limitations observed in the "hashing-enhanced" learning strategy, our study here is primarily motivated by equation-based sketches (Fu et al., 2020; Sheng et al., 2021; Huang et al., 2021), from the perspective of compressed nature of sketching algorithms (Cormode, 2011). Specifically, they employed a compressive sensing (CS) approach in the *query phase* to achieve a very low relative error, given counter values and per-key aggregations. While these works showed great applications of the CS theory (Cormode & Muthukrishnan, 2005a) to sketch-based estimation, their sensing matrix constructing and iterative-optimization-style recovery operation to get all frequencies of observed keys introduces considerable time and memory complexity even with state-of-the-art numerical solvers (Dalt et al., 2022).

Consequently, one cannot help but pose the following intriguing question: *Can we design a learning-based sketch (without ground truth) via the linear system by training and recovering on the fly?*

The comparison between the question-oriented approach in this work and existing learned sketches is shown in Fig. 1. Without slowing down the conventional sketch insertions, our approach continuously trains the ML model to recover per-key frequencies using only sketch counters. Although promising, the process raises the following two challenges: (1) The first is *self-supervision*. Unsupervised learning, without access to real frequencies or labels, is crucial for overcoming the impracticalities associated with existing learning-based frequency estimation algorithms. (2) The second is *scalability* or *complexity*. Many modern streaming scenarios have evolved into complex systems featuring tens of thousands of distinct items, and the entire key space invariably includes some uncertain keys that will be observed in the future. To decrease the complexity for per-key prediction, models dealing with such data need to be highly scalable as the size of streams grows infinitely.

To address these challenges, we introduce a new frequency estimation framework called **UCL-sketch** (**U**nsupervised **C**ompressive **L**earning **Sketch**), which aims to integrate the advantages of both equation-based and learned sketching approaches. This framework achieves a significant improvement in practical feasibility and accuracy compared to learning-augmented algorithms, while maintaining substantially lower query overhead than equation-based competitors. A notable distinction from prior works is that our model is entirely ground-truth free, relying solely on downsampled frequencies for online training. This property endows UCL-sketch with great flexibility and the capability for quick response to streaming distribution drift. To realize these benefits, we theoretically and empirically demonstrate that the recovery function can be learned from compressed measurements alone using an *equivalent learning* scheme, given per-key aggregations and keys set. Additionally, to mitigate the impact of large-scale and unbounded streams, we adopt the concept of *logical buckets* to split and jointly learn multiple bucket-associated mappings with shared parameters, leading to an efficient and expandable architecture. Experimental results demonstrate the potential of our proposed algorithm through detailed evaluations of the frequency estimation problem. Our contributions can be summarized as follows:

Figure 2: **The overall processing framework of equation-based sketch:** In the data plane, it builds a local sketch to record the data stream and a key tracking mechanism for new item identification and reporting. After the centralized server receives sketch counters and keys from the monitor device, the control plane can recover the frequencies through solving an under-constrained equation system.

- We open up a new direction of learning-based frequency estimation algorithm design. Specifically, we propose a more practical framework with learning technologies, dubbed as UCL-sketch, to recover per-key frequencies from compressed counters in the sketch, which is scalable, accurate, and self-supervised.
- We further provide a theoretical performance analysis of the ULC-sketch, and present how our training scheme enables solving data sketching problems without ground truth.
- We conduct an extensive evaluation with real-world and synthetic datasets to show that the proposed sketching method brings noticeable performance gain over existing state-of-the-art sketches.

## 2 PRELIMINARIES

### 2.1 KEY IDEAS OF EQUATION-BASED SKETCH

We follow prior studies that uncover the linear compression nature of equation-based sketch framework (Fu et al., 2020; Sheng et al., 2021; Huang et al., 2021). Typically, a sketching algorithm comprises an *insertion* component that feeds the key-value input to a compact structure that approximates these key-value pairs with one or multiple hash-based buckets arrays, a *recovery* component that inverses queried pairs from key-aggregations based on the same set of hash functions. Theorem 1 establishes the equivalence between the sketch with the linear system as follows:

**Theorem 1.** *The goal of frequency estimation based on a linear sketch is equivalent to solving linear equations from the given keys, hash functions, and counters. Let $x \in C^N$ denote the vector of the streaming key-frequency sequence and $y \in C^M$ denote sketch counters, the insertion process corresponds to $y = Ax$, while the result of recovery phase corresponds to*

$$x = A^{\dagger}y + (I - A^{\dagger}A)x, \tag{1}$$

*where $A \in C^{M \times N}$ is an indicator matrix of mapping the vector $x$ to a buckets array $y$, and $A^{\dagger} \in C^{N \times M}$ satisfies $AA^{\dagger}A \equiv A$.*

The proof can be found in Appendix C. As shown in Fig. 2, the equation-based sketch needs to build a local sketch like CM-sketch in the data plane and perform its original update operation. It deploys an additional key tracking mechanism to identify new keys and transfer them to the control plane. What distinguishes the design from other sketches is that it leverages an equation-based approach to compensate for per-key error caused by counter sharing in the recovery phase (or control plane), which can be concluded as three steps: (i) Transform sketch counters to the measurement vector $y$. (ii) For all distinct stream items, construct a sketch operator $A$ based on the hash functions that map them in the sketch. (iii) Fix the system of linear equations by an equation solver.

### 2.2 PROBLEM STATEMENT

Simply speaking, we formulate our frequency estimation algorithms via compressive sensing (CS), like the equation-based sketch introduced in Section 2.1. Let a data stream of running length $n$ be a sequence of $n$ tuples. The $t$-th tuple is denoted as $(k_t, v_t)$, where $k_t$ is a data-item key used for hashing and $v_t$ is a frequency value associated with the item. For each item, the insertion process applies an update to sketch counters (vector) $y$ of length $M$, with a sensing matrix $A$ defined by

hash functions and the key value as shown in Fig. 11. Then given the collected $y$ and $\boldsymbol{A}$, the output of recovery phase is a list of estimated frequencies, i.e., ground-truth (GT) vector $x$. Also note that we assume the size of possible keys space $N > M$ in this work, because keys are usually drawn from a large domain (e.g, IP addresses, URLs) while available space in data plane is limited, leading to an ill-posed system. Formally, the recovery phase builds an optimization problem: $\max_{x} \log p\left(x|y\right)$, s.t. $y = \boldsymbol{A}x$.

### 2.3 MOTIVATION

**Limitation of Baseline Solution.** There have been many implementations and extensions of equation-based sketch. Among these sketching solutions, the PR-sketch (Sheng et al., 2021) and SeqSketch (Huang et al., 2021) represent the most recent examples. These methods involve key tracking mechanisms to collect distinct keys in the stream and apply optimization techniques, such as Orthogonal Matching Pursuit (OMP), to solve a linear system. However, the computational cost of streaming problems grows significantly with the number of keys, making iterative process of "decode" algorithms less feasible for modern streams. As a result, they have remarkably increased per-key accuracy as well as computation time and peak memory consumption at query time.

**Impact of a Learned Equation Solver.** One idea here to eliminate the need for iterative optimization is a learned equation solver that directly maps measurement $y$ to frequencies $x$. Clearly, the deep solver requires extra training cost (but in parallel with stream processing), however, in return, it can greatly reduce the query processing time through one-shot prediction, while inheriting impressive performance of baseline solutions, just as they have done in the field of CS.

**Challenges of Supporting Learning-based Solution.** Building upon the above discussions, our primary goal is to develop equation-based sketches that exploit learning a solver to automate the process of per-key recovery. However, there are two key challenges: (1) Ensure online training without any ground truth, since it is very difficult to collect the true frequencies of all possible keys in real time for training; (2) Alignment with large-scale data stream under limited parameters overhead as the solver's complexity increases monotonically over time. In what follows, we give a detailed description of the proposed self-supervised learning framework for per-key recovery.

## 3 METHODOLOGY

In this section, we present UCL-sketch for stream frequency estimation, and elaborate on its design.

### 3.1 BASIC DESIGN

**Key Ideas.** To mitigate the extra bandwidth overhead used for transmitting keys, the UCL-sketch employs a Bloom Filter during the update phase, ensuring that each unique key is identified and reported at most once. Moreover, we filter hot keys twice, storing them separately in a hash table and an array, which has been proven beneficial for skewed streams (Huang et al., 2021).

**Data Structure**. Fig. 3 depicts the data structure of UCL-sketch. In the data plane, it has three types of data structures: (1) a heavy filter (hash table) *HF* to track frequent key pairs, (2) a sketch to record the remaining items, and (3) a Bloom Filter *BF* for key identification. Each slot in the HF consists of three fields. In addition to a key identifier, the slot contains two counters: *new count*, which tracks the values associated with the key, and *old count*, which records the values not attributed to the key. For

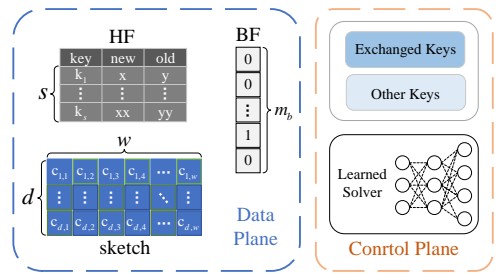

Figure 3: Data Structure of UCL-sketch.

the control plane, apart from a learned solver, we maintain two non-repeating and non-overlapping arrays to record inserted keys and exchanged keys from HF in our sketch, respectively.

**Update Operation**. The procedure of inserting an item in our UCL-sketch is very similar to previous equation-based sketch, e.g. SeqSketch (Huang et al., 2021). The main difference is the exchanged keys that are supposed to be relatively hot are stored separately in preparation for subsequent training phase. Due to space limitations, detailed descriptions are in Appendix B.

### 3.2 TRAINING STRATEGY

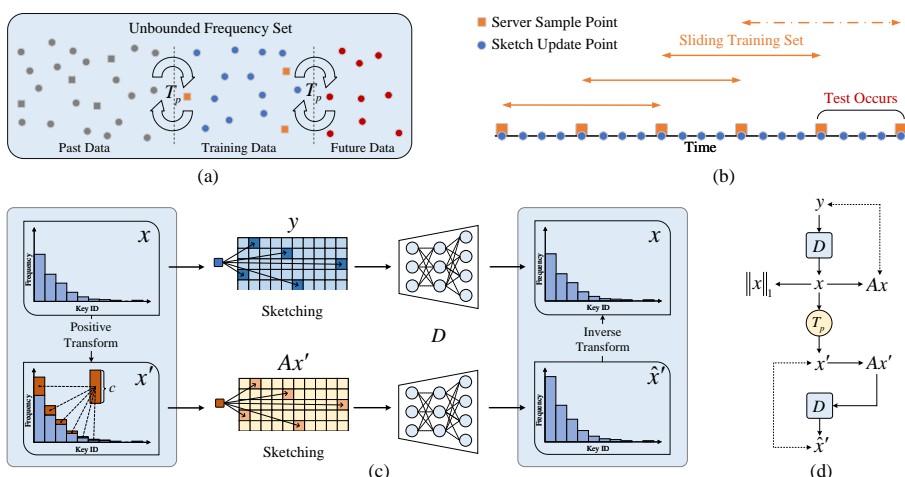

(a)            (b)

(c)            (d)

Figure 4: **Main ideas of training strategy:** (a) The unbounded set of frequency vectors is tolerant of certain Zipfian transformations. (b) Continually adapting the model using sampled counters in a sliding window for online training. (c) The learned solver $y \to x$ should also be invariant to these natural transformations. (d) Illustration of our self-supervised equivalent loss design.

**Goal and intuitions.** As mentioned in Section 2.3, we consider a challenging but reasonable setting in which only sampled measurement vector $y$, and the sketch sensing matrix $\boldsymbol{A}$ are available for on-line training our solver $D : y = \boldsymbol{A}x \to x$. As shown in Eq. 1, the root problem is a non-trivial null space defined by $(I - \boldsymbol{A}^{\dagger}\boldsymbol{A})x$ while $y$ only provides the information of range space of $\boldsymbol{A}$, i.e., its pseudo-inverse $\boldsymbol{A}^{\dagger}$. Therefore, a simple solver without additional constraints is not efficient enough to resolve the GT ambiguity. Our intuition for achieving unbiasedness is the distribution of item frequencies follow approximate Zipf's law, then the GT domain (output of the solver) should be invariant to certain groups of transformations that allow us to learn beyond the range space.

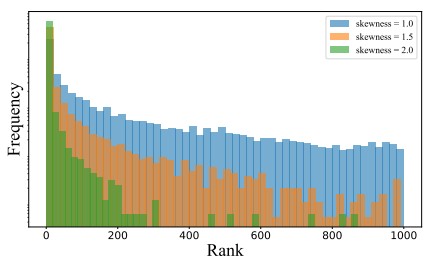

Figure 5: Example of the Zipfian distribution with different skewness.

**Online Training.** First of all, we present an online training procedure for continually adapting the learned solver. When analyzing or processing continuous streaming data, one only needs to keep track of recent stream because queries always occur in the future, while the useful information contained in past streaming data is diminishing over time. Therefore, as shown in Fig. 4 (b), our approach for dealing with non-stationary data streams is to adopt a sliding window (SW) mechanism which retains a fixed number of sampled "snapshots" of sketch counters in memory, instead of training on the entire history. The concept revolves around maintaining a "window" that slides with time, capturing only the recent state of the unbounded stream. The sample point depends on the times of sketch updates, for instance, these counters are transmitted to the control plane after every 1,000 insertions.

**Equivariant Learning.** A straightforward practice of unsupervised frequency recovery is to impose the measurement consistency on the model, using a range space loss of form like $\|\boldsymbol{A}x - y\|_2^2$. However, the sketch operator $\boldsymbol{A}$[1] has a null space, which means the model converges freely to the biased solution $\boldsymbol{A}^{\dagger}y + H(y)$ with $\boldsymbol{A}H(y) = 0$. This will cause unstable reconstructions without meeting ground truth data or prior information. According to CS theory, one direct way to alleviate the problem is "$L_1$ minimization" by adding a regularization $\|x\|_1$ penalizes against the lack of sparsity (Huang et al., 2021). Fortunately, it makes sense in streaming algorithms because heavy-tailed data such as network traffic exhibits high sparsity, but it is not enough to eliminate the impact of null space, since the accuracy of particular items is still unsatisfactory (Li et al., 2023).

---

[1]Note that we do not need to explicitly maintain the 0-1 sparse sensing matrix $\boldsymbol{A}$ in memory, although the control plane has sufficient space. Instead, we generate its elements on demand.

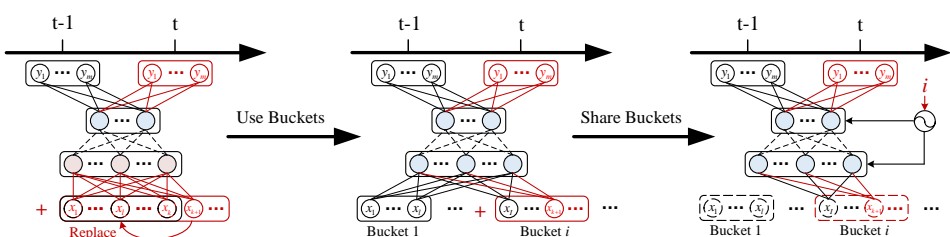

Figure 6: **Expansion design of a learned solver: Left:** *Redesigning*. Whenever a new key emerges, the entire output layer learned on previous streams is replaced and retrained. **Center:** *Partial-redesigning with buckets*. The keys are separated into independent buckets, so the solver is affected only by newly updated bucket. **Right:** *Non-redesigning with logical buckets*. By sharing buckets, the solver can be adapted to varying key space, and greatly reduces the number of parameters.

In order to learn more knowledge beyond the range space of $\boldsymbol{A}$, we draw on ideas from the Zipf Law prior of streaming item frequencies, and this is a common and natural reoccurring pattern in real-world data (Aamand et al., 2024). At their simplest, zipfian models are based on the assumption of a simple proportionality relationship: $f(k_j) \propto \frac{1}{j}$, where $f(k_j)$ is the frequency of $j$-th key $k_j$ in keys set sorted by volume as shown in Fig. 5. Furthermore, we posit an additional assumption of "temporal smoothness" in the heavy-tailed prior, observing that frequencies within adjacent temporal sample points tend to express similar zipfian behavior: a few frequent items exhibit approximate proportional growth, while other infrequent items remain their original size.

Under this mild prior information, we define a group of transformations $\mathcal{P} = \{p_{1,...,|\mathcal{P}|}\}$, in which arbitrary $p_i$ can be summarized in the following steps: Given a positive integer $c$ which usually takes value around the sampling interval, $p_i$ allocates it proportionally based on the volume of exchanged (or hot) keys in the input frequency vector. Next, the transformation randomly selects a small number of frequencies, i.e. $5\%$ from the remaining non-hot keys, and increments them by minimum update unit. As Fig. 4 (a) shows, for all possible $x$ in the unbounded frequency set $\mathcal{X}$, the equivalent relationship $T_p x \in \mathcal{X}$, $\forall p \in \mathcal{P}$ holds, where $T_p \in \mathbb{R}^{N \times N}$ is the corresponding transformation matrix of $p$. Then, our learned solver $D$ should also capture such invariant property of the target domain, that is, $D(\mathbf{A}T_p x) = T_p D(\mathbf{A}x)$. This additional constraint on the mapping allows the model to learn beyond the range space (see details in Theorem 3). If the incremental component can be handled by the solver, then the ambiguity in null space recovery can be effectively mitigated during learning. As shown in Fig. 4 (c) and (d), the network weights are updated by minimizing the following objective:

$$\arg\min_{\theta} \mathbb{E}_{y \in \boldsymbol{A}\mathcal{X}, p \in \mathcal{P}} \left\{ \|\boldsymbol{A}D(y) - y\|_2^2 + \lambda \|D(y)\|_1 + \|D(\boldsymbol{A}T_p D(y)) - T_p D(y)\|_2^2 \right\}, \quad (2)$$

where the first term enforces measurement consistency, the second term imposes sparse constraint, and the third term enforces system equivariance, and $\lambda$ is a trade-off coefficient. The training pseudo-code in one epoch is exhibited in Algorithm 1.

| **Algorithm 1** One-Epoch Training Algorithm of UCL-sketch | **Algorithm 2** Query Operation on the UCL-sketch |
|---|---|
| **Require:** $\eta$, $\lambda$, sketch matrix $\boldsymbol{A}$, exchanged key ids $h$, measurement set $\boldsymbol{Y}$, and number of keys $n$ | **Require:** learned model $D$, keys set $\boldsymbol{\Omega}$, key $k$, current measurement vector $y$, and bucket length $L$ |
| **Ensure:** trained model $D$ | **Ensure:** estimated frequency $x_k$ |
| 1: **for** measurement vector $y$ in $\boldsymbol{Y}$ **do** | 1: position $\leftarrow$ Get_Key_Position($\boldsymbol{\Omega}$, $k$); |
| 2:     $x \leftarrow$ Per_Key_Recovery($D$, $y$, $n$); | 2: bucket_id $\leftarrow$ position // $L$; |
| 3:     $x' \leftarrow$ Positive_Transform($x$, $h$, $n$); | 3: inner_id $\leftarrow$ position - bucket_id $\times L$; |
| 4:     $\hat{x}' \leftarrow$ Per_Key_Recovery($D$, $\boldsymbol{A}x$, $n$); | 4: $x \leftarrow D(y, \text{bucket\_id})$; |
| 5:     Take gradient descent step on | 5: $x_k^s \leftarrow x[\text{inner\_id}]$; |
|         $\eta\nabla_{\theta_D}(\|\boldsymbol{A}x - y\|_2^2 + \|\hat{x}' - x'\|_2^2 + \lambda\|x\|_1)$; | 6: $x_k \leftarrow x_k^s +$ Heavy_Filter_Query($k$); |
| 6: **end for** | 7: **return** $x_k$; |
| 7: **return** $D$; | |

## 3.3 SCALABLE ARCHITECTURE

**Goal and intuitions.** UCL-sketch is explicitly designed to allow for sketching large-scale data steam, where an unknown number of new items arrive at the monitor device in sequence, rather than designed for a fixed or small key space. Specifically, our goal here is to let the learned solver

Figure 7: The overview of complete version of our UCL-sketch.

dynamically expand its capacity once new elements arrive, while achieving efficiency in parameters. The intuition for designing such a lifelong network is to incrementally adapt to new items while retaining acquired frequencies of previous items, so parameter sharing is a good and natural choice.

**Network Expansion**. A most naive way to design the network for a sequence of items would be retraining the output layer(s) every time a new item emerges. However, such retraining would incur significant costs for a deep neural network. Instead, we suggest dividing keys set into small buckets, where each bucket is maintained with its own parameters, thereby reducing the extra expanding overhead. Given the information collected about the observed stream, this design transforms the problem into a maximum-a-posteriori (MAP) estimate of the bucket-associated frequencies (Dalt et al., 2022). Unfortunately, it is still very intensive in terms of memory usage since the network's size scales with the observed keys, under high-speed streams where the computational cost is a significant concern. Note that the statistical properties of sketch-based estimation should be stationary over buckets, as it implies that the similar underlying transformation (random hash) can be applied at each key (or bucket) inserted in the sketch. Thus, a better solution in such a case is sharing parameters across these buckets. Specifically, we train a solver to model the mapping $(y, i) \rightarrow x_i$, in which $i$ denotes the index of our selected bucket and $x_i \subset x$ is predicted frequencies in the logical bucket. Borrowing ideas from literature in diffusion models (Song et al., 2020; Ho et al., 2020), we learn the bucket-shared network using the sinusoidal position embedding (Vaswani et al., 2017). After this bucket sharing and logification, the network needs to update its weights by repeatedly forward propagation since the split changes the overall structure. Fortunately, in practice, this operation can be performed for all logical buckets in parallel. Fig. 6 illustrates our dynamically scalable network architecture and shows what happens when the set is partitioned into independent buckets.

**Query Operation**. When querying an item $k$, we initially locate its index in the keys set such that we can determine the bucket id and relative position of $k$. Then we query the hash table in the data plane to obtain its filtered frequency, and return 0 if the key is not in it. The partial result and sampled sketch counters will be reported to the server together. With the previously acquired position, we predict the remaining frequency of $k$ by inputting counters and bucket id into the learned solver. The final estimated frequency is a sum of the two parts. The process is depicted in Algorithm 2.

### 3.4 Putting It Together

We now put the basic design and our optimizations together, to build the final version of UCL-sketch. Fig. gives overview of the complete version. Fig. 7 (a) shows the procedure of insertion process in data plane. Then in Fig. 7 (b) and (c), we construct the corresponding sketch sensing matrix $A$ and present scalable inference of our bucket-wise network, respectively. To train the deep solver, since no real frequency is sent to the control plane, as illustrated in Fig. 7 (d), our goal is to reconstruct per-key frequencies $x$ conforms to three constraints: measurement, sparsity and Zipfian distribution. Finally, to query the frequency of keys, we sum up the results from the learned solver and the hash table as shown in Fig. 7 (e).

## 4 Theoretical Analysis

In this section, we analyze the proposed UCL-sketch, including complexity, keys coverage, error bound, and requirement for unbiased estimation during training. Due to space constraints, we only list the conclusions here. Detailed proofs, remarks and bound comparison with prior works can be found in Appendix C.

**Notation**. We define the following notations. In the context of sketching algorithms: $\varepsilon$ controls the accuracy of the sketch; smaller $\varepsilon$ means higher accuracy but potentially larger sketch size. $\delta$ controls

the confidence of the result; smaller $\delta$ means higher confidence that the error is within bounds. Then given parameters $(\varepsilon_c, \delta_c)$, set sketch array width $w = \lceil e/\varepsilon_c \rceil$ and depth $d = \lceil \ln(1/\delta_c) \rceil$ where $e$ is the base of the natural logarithm, with cutoff ($s$ reserved slots equipped with $b$ flag bits for each pair) in the heavy filter. We use $(\varepsilon_b, \delta_b)$ as the coefficient and error probability of the Bloom Filter: the number of bits $m_b = k_b K / \ln 2$, $K$ is the number of true existing keys, and the number of hash functions $k_b = \log(1/\varepsilon_b \delta_b)$; $x_T := \{x(i) : i \in T; 0 : i \notin T\}$ where $x$ is a vector has the same size with $x_T$.

**Definition 1** *(s-restricted Isometry Constant (Candes & Tao, 2005)). For every integer $s = 1, 2, \ldots$, we define the s-restricted isometry constants $\sigma_s$ of a matrix $A$ as the smallest quantity such that*

$$(1 - \sigma_s) \|x\|_2^2 \leq \|Ax\|_2^2 \leq (1 + \sigma_s) \|x\|_2^2 \tag{3}$$

*for all s-sparse vectors, where a vector is said to be s-sparse if it has at most $s$ nonzero entries.*

Lemma 1 shows the complexities of memory space, and update time of UCL-sketch.

**Lemma 1**. *The space complexity of UCL-sketch is* $O\left(\frac{K}{\ln 2} \log \frac{1}{\varepsilon_b \delta_b} + \frac{e}{\varepsilon_c} \ln \frac{1}{\delta_c} + bs\right)$, *and the time complexity of update operation is* $O\left(\log \frac{1}{\varepsilon_b \delta_b} + \ln \frac{1}{\delta_c}\right)$ *in the data plane.*

Lemma 2 guarantees the error bound for missing keys in the recovery phase of UCL-sketch.

**Lemma 2**. *The keys coverage of the Bloom Filter in UCL-sketch obeys*

$$\Pr(Y \geq K_y) \leq \frac{K}{K_y}\left(1 - e^{-\frac{k_b K}{m_b}}\right), \tag{4}$$

*where variable $Y$ denotes the number of keys that are not covered but viewed as covered.*

We show UCL-sketch's worst-case error bound of per-key recovery (without equivalent loss) from sketch counters as shown in Theorem 2.

**Theorem 2**. *Let $f = (f(1), f(2), \ldots, f(n))$ be the real volume vector of a stream that is stored in the sketch, where $f(i)$ denotes the volume of $i$-th distinct item. Consider $T_0$ as the locations of the $s$ largest volume of $f$, and $T_0^c$ as the complement of $T_0$. Assume that the reported volume vector $f^*$ is the optimal solution that minimizes the first two objective in Eqn. 2. Then the worst-case frequency estimation error is bounded by*

$$\|f^* - f\|_1 \leq \frac{2 + (2\sqrt{2} + 2)\sigma_{2s}}{1 - (\sqrt{2} + 1)\sigma_{2s}} \|f_{T_0^c}\|_1 \tag{5}$$

Theorem 3 gives the necessary condition of UCL-sketch to achieve full accuracy in the training set with our GT-free learning strategy.

**Theorem 3**. *A necessary condition for recovering the true volume from compressed counters is that the following linear system has a unique solution:*

$$Bx = \begin{pmatrix} A \\ AT_{p_1} \\ \vdots \\ AT_{p_{|\mathcal{P}|}} \end{pmatrix} x = \begin{pmatrix} y \\ y^1 \\ \vdots \\ y^{|\mathcal{P}|} \end{pmatrix}, \tag{6}$$

where $y^{(\cdot)}$ is the measurement corresponding to the transformation $p_{(\cdot)}$. Rigorously, $\text{rank}(B) = n$.

## 5 EVALUATION

We next report key results compared to state-of-the-art methods with real-world steam datasets.

### 5.1 EXPERIMENTAL SETUP

**Datasets.** We conduct experiments on three real-world datasets. The first is *CAIDA* (Caida), real traffic data collected on a backbone link between Chicago and Seattle in 2018. We form 13-byte keys

with five fields: source and destination IP addresses, source and destination ports, and protocol. In our experiments, we use 1 million packets of it with around 100K distinct keys. The second is *Kosarak* (Ferenc), consists of anonymized click-stream data from a Hungarian online news portal. We also extract a segment of data with a length of 1 million for our experiments, where about 25K unique keys are in it. The third is *Retail* (Brijs et al., 1999), which contains retail market basket data supplied by an anonymous Belgian retail supermarket store. There are nearly 910K items in this stream, and we utilize the entire dataset comprising a total of 16K unique keys. Each key is 4-byte long in the above two datasets.

**Metrics.** For comparing the accuracy of frequency estimations, we leverage *Average Absolute Error (AAE)* and *Average Relative Error (ARE)*. Additionally, we use *Weighted Mean Relative Difference (WMRD)* and *Entropy Relative Error* to evaluate the accuracy of the per-key distribution. The detailed descriptions can be found in Appendix D.1.

**Algorithm Comparisons.** For comparison, we implement nine existing frequency estimation algorithms, such as *CM-sketch (CM)* (Cormode & Muthukrishnan, 2005b), *C-sketch (CS)* (Charikar et al., 2002), *Elastic Sketch (ES)* (Yang et al., 2018a), *UnivMon (UM)* (Liu et al., 2016), *Nitrosketch (NS)* (Liu et al., 2019) and three ideally learned sketches: *Learned CM-sketch (LCM)*, *Learned C-sketch (LCS) (Hsu et al., 2019)* and *Leaned Sketch (LS) (Aamand et al., 2024)*. Two baselines using zeros and means as estimates are also provided, denoted by *0s* and *Ms*, respectively. In addition, we compare our UCL-sketch with SeqSketch (Huang et al., 2021), an equation-based sketch, in our ablational experiments. We give them the same local memory of buckets & sketch as ours and present their formal details in Appendix D.4.

## 5.2 PERFORMANCE COMPARISON

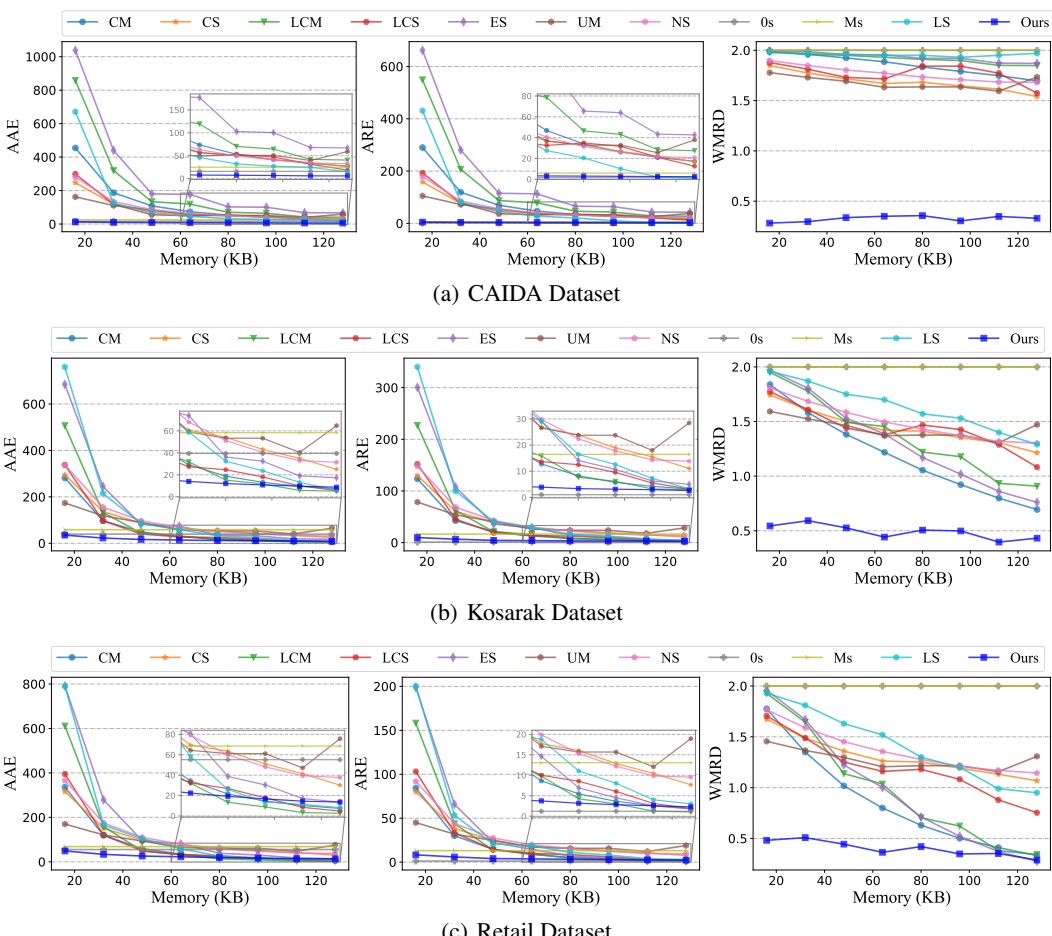

Figure 8: Performance comparison between our UCL-sketch and existing state-of-the-art sketches.

**Estimation Results.** First, we compare the AAE, ARE, and WMRD metrics for all seven sketches, by varying the local memory budgets from 16KB to 128KB. In the first two columns of Fig. 8, all methods achieve smaller AAE and ARE by increasing the space budget, whereas the UCL-sketch nearly always performs the best. In particular, when the memory cost is lower than 64KB, UCL-sketch achieves 6∼20 times smaller error rate, compared to the best algorithm. On the stream data with much larger key space, i.e. CAIDA dataset, there is a more remarkable accuracy gap between our method and other sketches in all memory settings. For example, AREs of CM, CS, LCM, LCS, ES, UM, NS are 23.77 times, 15.67 times, 41.39 times, 16.45 times, 54.04 times, 14.78 times, and 17.25 times of that of ours on average, respectively. Despite the additional BF to keep tracking unique items, the reason is that existing algorithms cannot estimate per-key aggregations accurately without leveraging the linear system of sketching.

Regarding frequency distribution alignment, we find that UCL-sketch still achieves better accuracy than the state-of-the-art sketches. As shown in the last column in Fig. 8, we observe that UCL-sketch maintains the WMRD value lower than 0.55 in all settings on three datasets. Unfortunately, such a stable performance does not persist in other competitors. We see that they are significantly less precise than UCL-sketch, although the metric drops as memory increases in general. Even worse, the baselines only achieve WMRDs over 1.5 with 128KB of memory on CAIDA, because the trace set contains too many keys to reliably measure the distribution of the stream with traditional point-wise methods, and their desired resources exceed the hardware capacity. Moreover, we list entropy relative errors of all estimation algorithms in Table 4 (see Appendix E). As expected, similar trends to WMRD on the frequency entropy can be observed in the table, where our algorithm consistently achieves the smallest relative error, substantially outperforming the second-best. Overall, UCL-sketch achieves both high performance in frequency query and distributional accuracy.

**Processing Speed.** We perform insertions of all items in a stream, record the total time used, and calculate the throughput. By using Kosarak with 64KB of memory, we evaluate the throughput of UCL-sketch and other solutions under different key sizes. Fig. 9 provides the comparison results of processing speed, in which we run a simple two-layer RNN model once before each update operation to simulate the actual practice of learning-augmented algorithms. From the figure, it becomes evident that previous learning-based sketch, i.e. LCM and LCS, fails to meet the requirements of processing high-speed data streams, since the propagation time overwhelms the insertion time, encumbering the efficiency of sketching. Note that LS improves the throughput via parsimonious learning, which only query the heavy-hitter oracle with some probability (we set 2% here). Nevertheless, their processing speed is still inferior to other methods. Meanwhile, ES achieves the highest throughput through its lightweight data structure. Fig. 9 also shows that ULC-sketch which performed as the second fastest can achieve almost 50× speed of learning-augmented sketch due to the model-free update and heavy filter in the data plane.

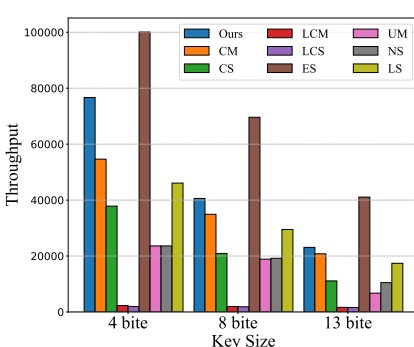

Figure 9: Processing speed comparison.

### 5.3 Additional Experiments

We put additional experiments on ablational study, synthetic zipfian data, parameter sensitivity and analysis which we omitted due to space limitations in the main body of the paper in Appendix F.

## 6 Conclusion

In this paper, we present the first GT-free learning-based frequency estimation algorithm, called UCL-sketch which provides a novel perspective for approximate measurement by leveraging the binding between sketch sensing and machine learning. Through extensive evaluation, the efficiency and accuracy of the UCL-sketch demonstrate the power of our methodology. Finally, we hope that this work will spark more research in the area of learning combinatorial sketching techniques.

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

# Appendix to "Toward Practical Learning-based Frequency Estimation without Ground Truth"

## A  RELATED WORK

**Classic Sketch**. A sketch is a compact structure and solution that takes limited space to support approximate frequency queries over high-speed data streams. Classic sketch algorithms (Cormode & Muthukrishnan, 2005b; Charikar et al., 2002; Estan & Varghese, 2002; Roy et al., 2016; Liu et al., 2016; Yang et al., 2018a; Liu et al., 2019) include Count-Min Sketch (CM-sketch), Count Sketch (C-sketch), Conservative Update Sketch (CU-sketch), Augmented Sketch (A-sketch), and so on. They adopt a common underlying structure which is essentially a $w \times d$ array of counters for preserving key frequencies. Each of the $d$ rows of the array is associated with a hash function for mapping items to $w$ counters, then they consider the counts of $d$ different buckets array (e.g. minimum for CM-sketch (Cormode & Muthukrishnan, 2005b), the median for C-sketch (Charikar et al., 2002)) to which the data is mapped as the estimation. The CU-sketch (Estan & Varghese, 2002) changed the insertion of the CM-sketch, which only updates the value of the minimum bucket in each insertion process. The A-sketch (Roy et al., 2016) added a filter to the CM-sketch and exchanged data between the filter and the sketch to ensure that hot keys are retained in the filter to reduce hash conflicts. Since classic sketches have been proven to deliver high accuracy only with impractical memory consumption, these algorithms are subject to an undesirable compromise between estimation accuracy and memory efficiency.

**Equation-based Sketch**. Recent works have made progress in mitigating the trade-off by designing advanced query methods of sketch algorithms. Lee et al. (2005) and Lu et al. (2008) first disclose that sketch and compressive sensing are thematically related. The locality-sensitive sketch (LSS) (Fu et al., 2020) leverages the relationship between the sketch with the compressed projection, then extends it to a K-means clustering method. The PR-sketch (Sheng et al., 2021) recovers keys of streaming data by establishing linear equations. The SeqSketch (Huang et al., 2021) and HistSketch (He et al., 2023) store a few high-frequency items, then employ a compressed-sensing approach to decode infrequent keys. By solving the linear system, these equation-based sketches compensate for the error introduced by counter sharing, and recovers the complete keys in the shared part with much higher accuracy than the classical sketch. Unfortunately, such global sketches have been shown to suffer from greatly increased time and memory costs of estimation.

**Learning-based Sketch**. In the last few years, machine learning has taken the world by storm than ever before, which also motivates the design of learning-based frequency estimation algorithms. Yang et al. (2018b) pioneered the idea of employing machine learning to reduce the dependence of the accuracy of sketches on network traffic characteristics. TalentSketch (Yan et al., 2022) applies a long short-term memory (LSTM) model to network measurement tasks. And most recently Cao et al. (2024) introduced meta-learning into sketch design, which can be robust to different local distributions, with performance scarcely affected by shifts in item-frequency correspondences. In (Hsu et al., 2019), the authors first proposed a learned frequency estimation framework by using a trained classifier (or oracle) to store hot and cold items separately. The overall design is similar to A-sketch, but the latter one uses a data exchange structure. Jiang et al. (2020) adjusted the learning-augmented sketch, which uses a regression model to directly outputs the predicted frequency of hot keys rather than inserting them in unique buckets. Then a series of works have also studied theoretical analyses and optimizations under this framework (Du et al., 2021; Cheng et al., 2023; Aamand et al., 2024). However, these hand-derived methods are excessively dependent on offline models and cannot handle dynamic data distribution. Differently, UCL-sketch provides a new paradigm for learning-enhanced sketch design. It obtains accuracy close to original equation-based sketches while maintaining the efficient query execution time by continuously adapting models without ground truth.

**Algorithm 3** Update Operation on the UCL-sketch

**Require:** key-value pair $(k, v)$
**Ensure:** inserted UCL-sketch
1: $i \leftarrow \mathrm{hash}\,(k)$
2: **if** HF[i].key = Null **then**
3:     HF[i].new $\leftarrow v$; HF[i].old $\leftarrow 0$; HF[i].key $\leftarrow k$;
4: **else**
5:     **if** HF[i].key = $k$ **then**
6:         HF[i].new $\leftarrow$ HF[i].new + $v$;
7:     **else**
8:         HF[i].old $\leftarrow$ HF[i].old + $v$;
9:         **if** (HF[i].new - HF[i].old) > 0 **then**
10:             insert sketch with (HF[i].key, HF[i].new);
11:             insert BF with HF[i].key;
12:             report hot key HF[i].key to the control plane;
13:             HF[i].new $\leftarrow v$; HF[i].old $\leftarrow 0$; HF[i].key $\leftarrow k$;
14:         **else**
15:             insert sketch with $(k, v)$;
16:             **if** $k \notin$ BF **then**
17:                 insert BF with $k$;
18:                 report cold key $k$ to the control plane;
19:             **end if**
20:         **end if**
21:     **end if**
22: **end if**
23: **return** ;

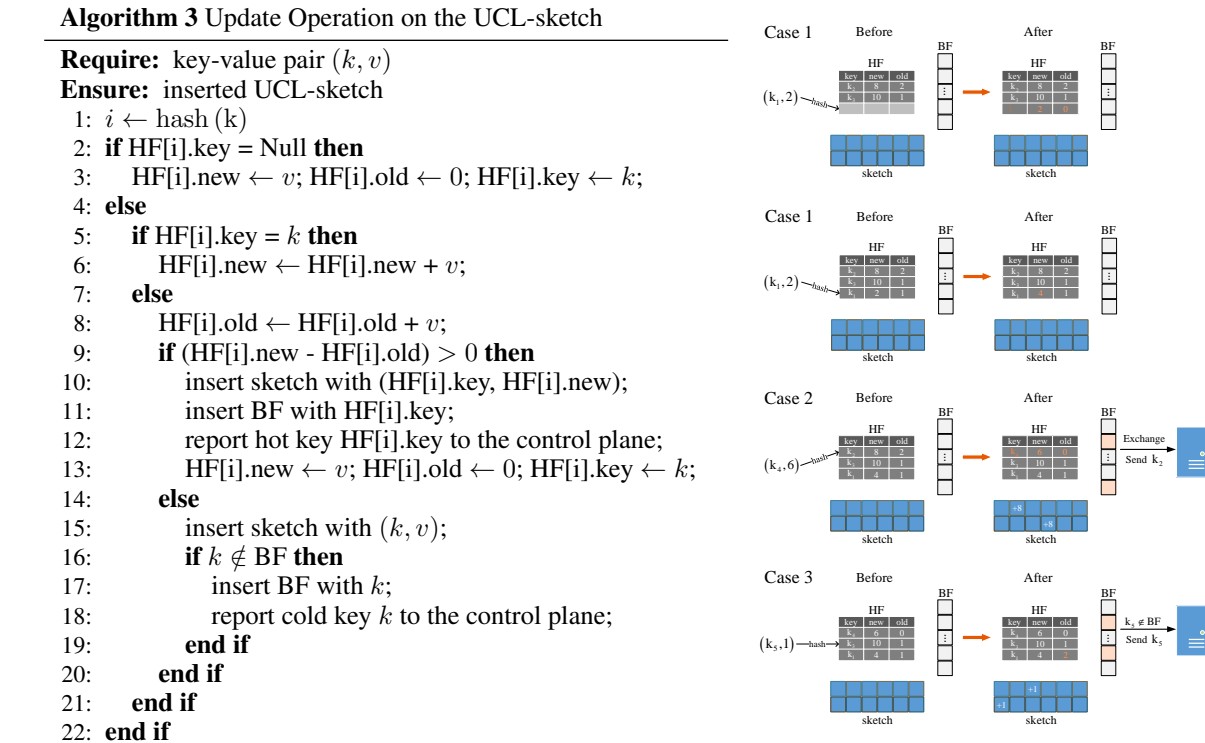

Figure 10: Examples of update processing.

**Compressive Sensing**. Given the linear measurements, Compressive Sensing (CS) methods usually reconstruct the original signal by solving an (generally convex) optimization problem, which is similar to ours. Many traditional works incorporate additional prior knowledge about transform coefficients (Candes et al., 2006; He & Carin, 2009; Kim et al., 2010; Zhao et al., 2016) into the CS reconstruction framework. Also note that the network-based CS reconstruction has been adopted in many magnetic resonance imaging (MRI) and super-resolution (SR) algorithms (Gregor & LeCun, 2010; Zhang & Ghanem, 2018; Pang et al., 2020; Chen et al., 2021; Xiao et al., 2023). However, UCL-sketch is the first to apply the idea to large-scale stream frequency estimation and specifically address the data sketching problem.

## B UPDATE ALGORITHM

Algorithm 3 outlines the procedure of inserting a key-value pair $(k, v)$. We first compute its hash position in HF, then as Fig. 11 shows, there are overall three cases:

*Case 1*: The slot is empty or the existing entry has the same key. We insert the item into the position or just increment the new count by its value.

*Case 2*: The current position does not have the same key and (new count - old count) $\leq 0$ after incrementing the old count by the item's value. We replace the existing entry with the new item and evict the old entry into the sketch. Then we transfer the exchanged key with the "hot" flag to the control plane, and update the BF.

*Case 3*: The current position does not have the same key and (new count - old count) > 0 after incrementing the old count by the item's value. We insert the item into the sketch. After inserting the BF, only if it is identified as a new key, UCL-sketch sends the key to the control plane.

Given a heavy-tailed distribution of stream data, Case 1 constitutes a large portion while Case 2 represents the opposite. Thus after filtering by the hash table, UCL-sketch is memory efficient as the number of exchanges is usually very small (Roy et al., 2016).

## C PROOFS AND REMARKS

In this section, we prove the theoretical results in the paper by partially following Candes (2008).

**Theorem 1.** *The goal of frequency estimation based on a linear sketch is equivalent to solving linear equations from the given keys, hash functions, and counters. Let $x \in C^N$ denote the vector of the streaming key-value sequence and $y \in C^M$ denote sketch counters, the insertion process corresponds to $y = \boldsymbol{A}x$, while the result of recovery phase corresponds to*

$$x = \boldsymbol{A}^\dagger y + (I - \boldsymbol{A}^\dagger \boldsymbol{A})x, \tag{7}$$

*where $\boldsymbol{A} \in C^{M \times N}$ is an indicator matrix of mapping the vector $x$ to a buckets array $y$, and $\boldsymbol{A}^\dagger \in C^{N \times M}$ satisfies $\boldsymbol{A}\boldsymbol{A}^\dagger \boldsymbol{A} \equiv \boldsymbol{A}$.*

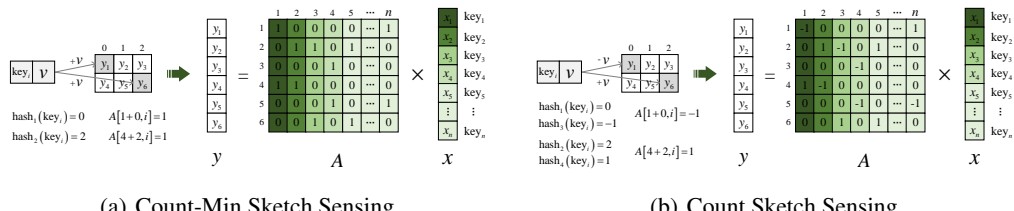

(a) Count-Min Sketch Sensing  (b) Count Sketch Sensing

Figure 11: The sensing process of and CM-sketch and C-sketch is based on the linear hash operation.

*Proof.* Suppose that the hashing operation randomly maps incoming items to a bucket array uniformly at random. For an incoming key-value pair, the sketch selects one counter indexed by hashing the key with a hash function at each array. Let $\boldsymbol{A}[i,j] = 1$ or $-1$ if the $j$-th key is mapped to the $i$-th bucket, and set other entries in this row vector to $0$s. Then the insertion process for all key-value pairs can be equivalently represented as an algebraic equation $y = \boldsymbol{A}x$. For example, we present the mapping matrices for CM-sketch and C-sketch in Fig. 11. Thus the approximated counters of a sketch can be calculated as a decoding phase: we can obtain the general solution of per-key $x$, i.e., $x = \boldsymbol{A}^\dagger y + (I - \boldsymbol{A}^\dagger \boldsymbol{A})t, \forall t \in C^N$, where the first part is in the range-space of $\boldsymbol{A}$ while the latter is in the null-space. One can justify this by left multiplying each side of the equation by $\boldsymbol{A}$. Then

$$x = \boldsymbol{A}^\dagger \boldsymbol{A}x + (I - \boldsymbol{A}^\dagger \boldsymbol{A})t \Leftrightarrow x - t = \boldsymbol{A}^\dagger \boldsymbol{A}(x - t) \tag{8}$$

Since $\boldsymbol{A}^\dagger \boldsymbol{A} \neq I$, we thus have $t = x$, which concludes the proof. $\square$

*Remark.* Considering Theorem 1, if we model a relation as defining a vector or matrix, then the sketch of this is obtained by multiplying the data by a (fixed) matrix. In this regard, a single update to the underlying volume has the effect of modifying a single entry in the frequency vector. Therefore, sketch-based frequency estimation is equivalent to solve a linear inverse problem.

**Lemma 1.** *The space complexity of UCL-sketch is $O\left(\frac{K}{\ln 2}\log\frac{1}{\varepsilon_b \delta_b} + \frac{e}{\varepsilon_c}\ln\frac{1}{\delta_c} + bs\right)$, and the time complexity of update operation is $O\left(\log\frac{1}{\varepsilon_b \delta_b} + \ln\frac{1}{\delta_c}\right)$ in the data plane.*

*Proof.* Since the bucket arrays of CM-sketch contain $w \times d$ counters, the total size of the heavy filter is fixed as $b \times s$, and the Bloom Filter occupies $m_b$ bits, the total space complexity in the data plane is $O(m_b + wd + bs) = O\left(\frac{K}{\ln 2}\log\frac{1}{\varepsilon_b \delta_b} + \frac{e}{\varepsilon_c}\ln\frac{1}{\delta_c} + bs\right)$. For each item in the stream, it first requires 1 hash operation to locate its slot in the heavy filter. Thus, the time complexity of filtering is only $O(1)$. Then the sketch hashes the key $d$ times, and the Bloom Filter hashes that $k_b$ times. Therefore, the time complexity of insertion is $O(k_b + d + 1) = O\left(\log\frac{1}{\varepsilon_b \delta_b} + \ln\frac{1}{\delta_c}\right)$. $\square$

**Lemma 2.** *The keys coverage of the Bloom Filter in UCL-sketch obeys*

$$\Pr(Y \geq K_y) \leq \frac{K}{K_y}\left(1 - e^{-\frac{k_b K}{m_b}}\right), \tag{9}$$

*where variable $Y$ denotes the number of keys which are not covered but viewed as covered.*

*Proof.* Suppose that independent hash functions uniformly map keys to random bits, the probability that a certain bit will still be 0 after one insertion is $1 - \frac{1}{m_b}$. Consequently, the probability that any bit of the Bloom Filter is 1 after $K_i$ distinct items have been seen is given by $1 - \left(1 - \frac{1}{m_b}\right)^{k_b K_i}$. We can use the identity $\left(1 - \frac{1}{m_b}\right)^{m_b} = \frac{1}{e}$ for large $m_b \to \infty$, then we have approximation $\left(1 - \frac{1}{m_b}\right)^{k_b K_i} \approx \left(\frac{1}{e}\right)^{k_b K_i / m_b}$. Therefore, the false positive probability of an unobserved key is $1 - e^{-k_b K_i / m_b}$. Also note that $E(Y) = \sum_{i=1}^{K} \left(1 - e^{-k_b i / m_b}\right) \leq K\left(1 - e^{-k_b K / m_b}\right)$. Now by Markov's inequality, the bound can be derived as: $\Pr\left(Y \geq K_y\right) \leq \frac{E(Y)}{K_y} \leq \frac{K}{K_y}\left(1 - e^{-k_b K / m_b}\right)$, which concludes the proof. $\square$

**Lemma 3.** *Given disjoint subsets $T_a, T_b \subseteq \{1, 2, 3, \dots\}$ with $|T_a|, |T_b| = s$, and $x$ an arbitrary vector can be supported on them, we have $\langle Ax_{T_a}, Ax_{T_b}\rangle \leq \sigma_{2s}\|x_{T_a}\|_2\|x_{T_b}\|_2$.*

*Proof.* The proof of Lemma 3 can be concluded from (Candes, 2008). According to (Candes, 2008), $\langle Ax, Ax'\rangle \leq \sigma_{s+s'}\|x\|_2\|x'\|_2$ holds for all $x, x'$ supported on the disjoint subsets. We replace $x, x'$ with $x_{T_a}, x_{T_b}$. Then, we have $\langle Ax_{T_a}, Ax_{T_b}\rangle \leq \sigma_{2s}\|x_{T_a}\|_2\|x_{T_b}\|_2$. $\square$

**Theorem 2.** *Let $f = (f(1), f(2), \dots, f(n))$ be the real volume vector of a stream that is stored in the sketch, where $f(i)$ denotes the volume of $i$-th distinct item. Consider $T_0$ as the locations of the $s$ largest volume of $f$, and $T_0^c$ as the complement of $T_0$. Assume that the reported volume vector $f^*$ is the optimal solution that minimizes the first two objective in Eqn. 2. Then the worst-case frequency estimation error is bounded by*

$$\|f^* - f\|_1 \leq \frac{2 + \left(2\sqrt{2} + 2\right)\sigma_{2s}}{1 - \left(\sqrt{2} + 1\right)\sigma_{2s}}\|f_{T_0^c}\|_1 \tag{10}$$

*Proof.* Given $f = f^* + l$, we start by dividing the residual vector $l$: let $T_1$ denote the locations of the largest $s$ value in $l_{T_0^c}$, $T_2$ denote the locations of the next largest $s$ value in $l_{T_0^c}$, and so on. By this definition, we can obtain $\left\|l_{T_j}\right\|_2 = \sqrt{\sum_i l_{T_j}^2(i)} \leq \sqrt{s \max_i^2\left(|l_{T_j}(i)|\right)} \leq \frac{1}{\sqrt{s}}\sum_i \left|l_{T_{j-1}}(i)\right| = \frac{1}{\sqrt{s}}\left\|l_{T_{j-1}}\right\|_1$. This holds for $j \geq 2$, which derives the following useful inequality:

$$\sum_{j\geq 2}\left\|l_{T_j}\right\|_2 \leq \frac{1}{\sqrt{s}}\sum_{j\geq 1}\left\|l_{T_j}\right\|_1 = \frac{1}{\sqrt{s}}\left\|l_{T_0^c}\right\|_1 \tag{11}$$

Since $f^*$ minimize $\|f^*\|_1$ subject to $\boldsymbol{A}f^* = \boldsymbol{A}f$, which intuitively means $\|f^*\|_1$ would not bigger than $\|f\|_1$, we have $\|f\|_1 \geq \|f^*\|_1$ and $\|\boldsymbol{A}f^* - \boldsymbol{A}f\|_2 = \|\boldsymbol{A}l\|_2 = 0$. It gives

$$\begin{aligned}
\|f_{T_0}\|_1 + \left\|f_{T_0^c}\right\|_1 = \|f\|_1 \geq \|f^*\|_1 &= \|f - l\|_1 = \left\|(f - l)_{T_0}\right\|_1 + \left\|(f - l)_{T_0^c}\right\|_1 \\
&\geq \|f_{T_0}\|_1 - \left\|f_{T_0^c}\right\|_1 + \left\|l_{T_0^c}\right\|_1 - \|l_{T_0}\|_1 \\
&\Leftrightarrow \left\|l_{T_0^c}\right\|_1 \leq 2\left\|f_{T_0^c}\right\|_1 + \|l_{T_0}\|_1.
\end{aligned} \tag{12}$$

Also note that $\|l_{T_0}\|_1 \leq \sqrt{s}\|l_{T_0}\|_2 \leq \sqrt{s}\|l_{T_{0\cup 1}}\|_2$, which is is derived by

$$\frac{1}{\sqrt{s}}\|l_{T_0}\|_1 = \sqrt{\left(\sum_i \frac{1}{\sqrt{s}}|l_{T_0}(i)|\right)^2} \leq \sqrt{\left(\sum_{j=1}^{s}\frac{1}{s}\right)\sum_i l_{T_0}^2(i)} = \|l_{T_0}\|_2 \tag{13}$$

using Cauchy–Schwarz inequality. And following from Definition 1, one can get $(1 - \sigma_{2s})\|l_{T_{0\cup 1}}\|_2^2 \leq \|\boldsymbol{A}l_{T_{0\cup 1}}\|_2^2$. Therefore, we instead bound $\|\boldsymbol{A}l_{T_{0\cup 1}}\|_2^2$ as follows:

$$\begin{aligned}
\|\boldsymbol{A}l_{T_{0\cup 1}}\|_2^2 &= \langle \boldsymbol{A}l_{T_{0\cup 1}}, \boldsymbol{A}l_{T_{0\cup 1}}\rangle = \langle \boldsymbol{A}l_{T_{0\cup 1}}, \boldsymbol{A}\left(l - l_{T_{0\cup 1}^c}\right)\rangle \\
&\leq \left|\langle \boldsymbol{A}l_{T_{0\cup 1}}, \boldsymbol{A}l\rangle\right| + \left|\langle \boldsymbol{A}l_{T_{0\cup 1}}, \boldsymbol{A}l_{T_{0\cup 1}^c}\rangle\right| \leq \|\boldsymbol{A}l_{T_{0\cup 1}}\|_2\|\boldsymbol{A}l\|_2 + \sum_{j\geq 2}\left|\langle \boldsymbol{A}l_{T_{0\cup 1}}, \boldsymbol{A}l_{T_j}\rangle\right| \\
&\approx 0 + \sum_{j\geq 2}\left|\langle \boldsymbol{A}l_{T_0} + \boldsymbol{A}l_{T_1}, \boldsymbol{A}l_{T_j}\rangle\right| \leq \sum_{j\geq 2}\left|\langle \boldsymbol{A}l_{T_0}, \boldsymbol{A}l_{T_j}\rangle\right| + \sum_{j\geq 2}\left|\langle \boldsymbol{A}l_{T_1}, \boldsymbol{A}l_{T_j}\rangle\right|.
\end{aligned} \tag{14}$$

Here using the Lemma 3, we have

$$\sum_{j \geq 2} \left| \left\langle \boldsymbol{A} l_{T_0}, \boldsymbol{A} l_{T_j} \right\rangle \right| \leq \sigma_{2s} \|l_{T_0}\|_2 \sum_{j \geq 2} \|l_{T_j}\|_2 \tag{15}$$

The proof for the upper bound of $\sum_{j \geq 2} \left| \left\langle \boldsymbol{A} l_{T_1}, \boldsymbol{A} l_{T_j} \right\rangle \right|$ follows from the similar procedure, and thus applying Ineqn. 11,

$$\|\boldsymbol{A} l_{T_{0 \cup 1}}\|_2^2 \leq \sigma_{2s} \left( \|l_{T_0}\|_2 + \|l_{T_1}\|_2 \right) \sum_{j \geq 2} \|l_{T_j}\|_2$$

$$\leq \sqrt{2} \sigma_{2s} \|l_{T_{0 \cup 1}}\|_2 \sum_{j \geq 2} \|l_{T_j}\|_2 \leq \sqrt{\frac{2}{s}} \sigma_{2s} \|l_{T_{0 \cup 1}}\|_2 \|l_{T_0^c}\|_1 \tag{16}$$

where the first part of the second line is derived by

$$\|l_{T_0}\|_2 + \|l_{T_1}\|_2 = \sqrt{\|l_{T_0}\|_2^2 + \|l_{T_1}\|_2^2 + 2\|l_{T_0}\|_2\|l_{T_1}\|_2} \leq \sqrt{2 \left( \|l_{T_0}\|_2^2 + \|l_{T_1}\|_2^2 \right)} = \sqrt{2} \|l_{T_{0 \cup 1}}\|_2 \tag{17}$$

Then recall that

$$\|l_{T_{0 \cup 1}}\|_2^2 \leq \frac{1}{1 - \sigma_{2s}} \|A l_{T_{0 \cup 1}}\|_2^2 \leq \frac{\sqrt{2/s}\sigma_{2s}}{1 - \sigma_{2s}} \|l_{T_{0 \cup 1}}\|_2 \|l_{T_0^c}\|_1$$

$$\Leftrightarrow \|l_{T_{0 \cup 1}}\|_2 \leq \frac{\sqrt{2/s}\sigma_{2s}}{1 - \sigma_{2s}} \|l_{T_0^c}\|_1 \tag{18}$$

which gives $\|l_{T_0}\|_1 \leq \sqrt{s}\|l_{T_{0 \cup 1}}\|_2 \leq \frac{\sqrt{2}\sigma_{2s}}{1 - \sigma_{2s}}\|l_{T_0^c}\|_1$. Now we combine it with Ineqn. 12 to obtain the certain bound

$$\|l_{T_0}\|_1 \leq \frac{\sqrt{2}\sigma_{2s}}{1 - \sigma_{2s}} \|l_{T_0^c}\|_1 \leq \frac{\sqrt{2}\sigma_{2s}}{1 - \sigma_{2s}} \left( 2\|f_{T_0^c}\|_1 + \|l_{T_0}\|_1 \right)$$

$$\Leftrightarrow \|l_{T_0}\|_1 \leq \frac{2\sqrt{2}\sigma_{2s}}{1 - \left(\sqrt{2} + 1\right)\sigma_{2s}} \|f_{T_0^c}\|_1. \tag{19}$$

Finally, the error bound of reported volumes is given by

$$\|f^* - f\|_1 = \|l\|_1 = \|l_{T_0^c}\|_1 + \|l_{T_0}\|_1 \leq 2\|f_{T_0^c}\|_1 + 2\|l_{T_0}\|_1 \leq \frac{2 + \left(2\sqrt{2} + 2\right)\sigma_{2s}}{1 - \left(\sqrt{2} + 1\right)\sigma_{2s}} \|f_{T_0^c}\|_1, \tag{20}$$

which concludes the proof. $\qquad\square$

Table 1: Theoretical comparison with state-of-the-art frequency estimation algorithms

| Algorithm | Space Complexity | Time Complexity | Expected Error | Reference |
|---|---|---|---|---|
| CM-sketch | $\mathcal{O}\left(\frac{e}{\varepsilon_c}\ln\frac{1}{\delta_c}\right)$ | $\mathcal{O}\left(\ln\frac{1}{\delta_c}\right)$ | $\mathcal{O}\left(\frac{\log(K)}{M}\right)$ or $\mathcal{O}\left(K\varepsilon_c\|F\|_1\right)$ | Hsu et al. (2019) |
| C-sketch | $\mathcal{O}\left(\frac{e}{\varepsilon_c^2}\ln\frac{1}{\delta_c}\right)$ | $\mathcal{O}\left(\ln\frac{1}{\delta_c}\right)$ | $\mathcal{O}\left(\frac{1}{M}\right)$ or $\mathcal{O}\left(K\varepsilon_c\|F\|_2\right)$ | Aamand et al. (2024) |
| Learned CM-sketch | $\mathcal{O}\left(S_o + \frac{e}{\varepsilon_c}\ln\frac{1}{\delta_c} + bs\right)$ | $\mathcal{O}\left(T_o + \ln\frac{1}{\delta_c}\right)$ | $\mathcal{O}\left(\frac{\log^2(K/M)}{M\log K}\right)$ | Hsu et al. (2019) |
| Learned C-sketch | $\mathcal{O}\left(S_o + \frac{e}{\varepsilon_c^2}\ln\frac{1}{\delta_c} + bs\right)$ | $\mathcal{O}\left(T_o + \ln\frac{1}{\delta_c}\right)$ | $\mathcal{O}\left(\frac{\log(K/M)}{M\log K}\right)$ | Aamand et al. (2024) |
| Elastic Sketch | $\mathcal{O}\left(\frac{-H\log K}{\ln(1-\varepsilon_b\delta_b)} + \frac{e}{\varepsilon_c^2}\ln\frac{1}{\delta_c}\right)$ | $\mathrm{O}\left(\ln\frac{1}{\delta_c}\right)$ | $\mathcal{O}\left(K\varepsilon_c\|f\|_1\right)$ | Yang et al. (2018a) |
| UCL-sketch | $\mathcal{O}\left(\frac{K}{\ln 2}\log\frac{1}{\varepsilon_b\delta_b} + \frac{e}{\varepsilon_c}\ln\frac{1}{\delta_c} + bs\right)$ | $\mathcal{O}\left(\log\frac{1}{\varepsilon_b\delta_b} + \ln\frac{1}{\delta_c}\right)$ | $\mathcal{O}\left(\|f_{T_0^c}\|_1\right)$ | Lemma 2 & Theorem 3 |

***Remark***. We list theoretical comparison between UCL-sketch and six methods in Table 1. Note that expected error $:= \sum_i \frac{f(i)}{\sum_i f(i)} \cdot |f^*(i) - f(i)| \leq \|f^* - f\|_1$. Here, we additionally define $H$ as the number of hot keys, $F$ as the unfiltered frequencies, $S_o$ and $T_o$ as the used memory and inference time of learned oracle in the learning-augmented sketch, respectively. Although some classic sketches have slightly lighter structure than ours, they obtain more inaccurate error bound. Specifically, it is worthy noting that $\|f_{T_0^c}\|_1 \ll K\varepsilon_c \|f\|_1$ in most scenarios due to heavy-tailed (sparse) property of real-world streams. It has also been proven in Yang et al. (2018a) that the bound

of Elastic Sketch is lower than that of C-sketch and CM-sketch, e.g., $\varepsilon_c \|f\|_1 < \varepsilon_c \|F\|_1$. Therefore, our equation-based algorithm significantly outperforms these three competitors regarding estimation accuracy. Meanwhile, the cost of learning-augmented sketches is heavily implicated by their learned oracle, while their error bounds stay coarse-grained. Note that the effect of equivalent learning was not considered in our analysis. Consequently, the practical performance achieved by our algorithms may surpass the theoretical result, which has been substantiated by the empirical results presented in Section 5.

**Theorem 3**. *A necessary condition for recovering the true volume from compressed counters is that the following linear system has unique solution:*

$$\boldsymbol{B}x = \begin{pmatrix} \boldsymbol{A} \\ \boldsymbol{A}T_{p_1} \\ \vdots \\ \boldsymbol{A}T_{p_{|\mathcal{P}|}} \end{pmatrix} x = \begin{pmatrix} y \\ y^1 \\ \vdots \\ y^{|\mathcal{P}|} \end{pmatrix}, \tag{21}$$

where $y^{(\cdot)}$ is the measurement corresponding to the transformation $p_{(.)}$. Rigorously, $\mathrm{rank}(\boldsymbol{B}) = n$.

*Proof.* Recall that the general form of the true frequency is $x = \boldsymbol{A}^\dagger y + (I - \boldsymbol{A}^\dagger \boldsymbol{A})x$ as Theorem 1 shows, thus for $p_1, p_2, \ldots, p_{|\mathcal{P}|} \in \mathcal{P}$,

$$x = \boldsymbol{A}^\dagger y + (I - \boldsymbol{A}^\dagger \boldsymbol{A})x$$
$$x = \left((\boldsymbol{A}T_{p_1})^\dagger y^1 + (I - (\boldsymbol{A}T_{p_1})^\dagger(\boldsymbol{A}T_{p_1}))\right) x$$
$$\vdots \qquad \vdots \qquad\qquad\qquad \vdots \tag{22}$$
$$x = \left((\boldsymbol{A}T_{p_{|\mathcal{P}|}})^\dagger y^{|\mathcal{P}|} + (I - (\boldsymbol{A}T_{p_{|\mathcal{P}|}})^\dagger(\boldsymbol{A}T_{p_{|\mathcal{P}|}}))\right) x.$$

Stacking all these equations together into

$$\begin{pmatrix} x \\ x \\ \vdots \\ x \end{pmatrix} = \begin{pmatrix} \boldsymbol{A}^\dagger y \\ (\boldsymbol{A}T_{p_1})^\dagger y^1 \\ \vdots \\ (\boldsymbol{A}T_{p_{|\mathcal{P}|}})^\dagger y^{|\mathcal{P}|} \end{pmatrix} + \begin{pmatrix} x \\ x \\ \vdots \\ x \end{pmatrix} - \begin{pmatrix} \boldsymbol{A}^\dagger \boldsymbol{A} \\ (\boldsymbol{A}T_{p_1})^\dagger \boldsymbol{A}T_{p_1} \\ \vdots \\ (\boldsymbol{A}T_{p_{|\mathcal{P}|}})^\dagger \boldsymbol{A}T_{p_{|\mathcal{P}|}} \end{pmatrix} x \tag{23}$$

By left multiplying each side of Eqn. 23 by $\left(\boldsymbol{A}, \boldsymbol{A}T_{p_1}, \cdots, \boldsymbol{A}T_{p_{|\mathcal{P}|}}\right)$, we have that

$$\boldsymbol{B}x = \begin{pmatrix} \boldsymbol{A} \\ \boldsymbol{A}T_{p_1} \\ \vdots \\ \boldsymbol{A}T_{p_{|\mathcal{P}|}} \end{pmatrix} x = \begin{pmatrix} y \\ y^1 \\ \vdots \\ y^{|\mathcal{P}|} \end{pmatrix}, \tag{24}$$

and therefore $\boldsymbol{B} \in \mathbb{R}^{|\mathcal{P}|m \times n}$ needs to be of full rank $n$ so that the true frequency $x$ can be accurately recovered from the null space.

***Remark***. Due to the set invariance of zipfian streaming data, Theorem 3 states the requirement to learn a deep solver without ground truth, is that multiple virtual sketch operators $\{\boldsymbol{A}T_{p_{(i)}}\}_{i=1,2,\ldots,|\mathcal{P}|}$ have enough different range space to determine unique per-key frequencies $x$. That means the choice of transformation group $\mathcal{P}$ in above system is not arbitrary, but needs to be rank $n$ or at least $> m$ so that the model is guaranteed to learn from the null space of $\boldsymbol{A}$. Critically though, much room beyond $\boldsymbol{A}^\dagger$ will be filled after a large number of random transformations during training. Thus, it is possible to train the solver from only sketch counters using our scheme, but also note that the group $\mathcal{P}$ might in some cases not be sufficient since the target $x$ is rapidly and continuously extending in complex streams.

# D  IMPLEMENTATION DETAILS

Our experiments run in a machine with one AMD 6-Core CPU (3.70 GHz), 32GB DRAM, and a single 12GB NVIDIA GeForce RTX 3060 GPU. For the learning-driven part, we used the PyTorch implementation. Besides, all these experiments are repeated multiple times using different fixed random seeds, and then their average results are reported in this paper.

## D.1 METRICS

There are five metrics used in our experiments:

(1) *Average Absolute Error (AAE)*: It equals $\frac{1}{n}\sum_{i=1}^{n}|f(i)-f^*(i)|$, where $f(\cdot)$ and $f^*(\cdot)$ are real and estimated frequency respectively.

(2) *Average Relative Error (ARE)*: It equals $\frac{1}{n}\sum_{i=1}^{n}\frac{|f(i)-f^*(i)|}{f(i)}$, where $f(\cdot)$ and $f^*(\cdot)$ are the same as those defined above.

(3) *Weighted Mean Relative Difference (WMRD)* (Sheng et al., 2021): It can be written as $\frac{\sum_{i=1}^{z}|n(i)-n^*(i)|}{\sum_{i=1}^{z}\frac{n(i)+n^*(i)}{2}}$, where $z$ is the maximum single-key frequency, and $n(i)$ and $n^*(i)$ are the real and estimated number of keys with frequency $i$ respectively.

(4) *Entropy Relative Error* (Yang et al., 2018a): We calculate the entropy $e$ based on a frequency set as $-\sum_{i=1}^{z}\left(i\times\frac{n(i)}{\sum_{i=1}^{z}n(i)}\log_2\frac{n(i)}{\sum_{i=1}^{z}n(i)}\right)$, then the relative error is $\frac{|e-e^*|}{e}$ where $e$ and $e^*$ are the true and estimated entropy. Here we define $0\log(0)=0$.

(5) *Throughput*: We use update operations per second to measure the processing speed of various sketching algorithms.

## D.2 PARAMETERS

Table 2: Parameter configurations of the local sketch and hash table

| Memory | 16KB | 32KB | 48KB | 64KB | 80KB | 96KB | 112KB | 128KB |
|---|---|---|---|---|---|---|---|---|
| Number of slots | 500 | 1500 | 2000 | 3000 | 3500 | 4500 | 5500 | 6000 |
| Depth of sketch | 4 | 4 | 4 | 4 | 6 | 6 | 6 | 8 |
| Width of sketch | 512 | 512 | 1024 | 1024 | 1024 | 1024 | 1024 | 1024 |

The parameters of the local sketch and hash table for each memory setting are listed in Table 2. As for the Bloom Filter, we determine the maximum number of bits by setting the coverage proportion over $99\%$ according to Lemma 2 and fixing $k_b$ as 8. We provide detailed parameters for the learned solver of this work in Table 3. We adopt the same key hyperparameter of NN throughout the experiments. In particular, $\lambda$ is the hyperparameter that reweights the sparse term in Eqn. 2. For optimization, we use Adam optimizer (Kingma & Ba, 2014) with default $(\beta_1, \beta_2)$ for all the experiments.

Table 3: hyperparameters setting and overhead for learning parts

| Hyperparameter | Setting value | Refer range |
|---|---|---|
| Bucket length | 512 | [128, 256, 512, 1024, 2056] |
| Extra layers | 0 | [0, 1, 2, 3] |
| Hidden dimension | 128 | [32, 64, 128, 256, 512] |
| Trade-off $\lambda$ | 0.1 | $0.05 \sim 1$ |
| Training epoch | 300 | $100 \sim 500$ |
| Patience | 30 | $10 \sim 50$ |
| Learning rate | 0.001 | $0.0001 \sim 0.01$ |
| Batch size | 32 | [8, 16, 32, 64, 128] |
| Sliding window length | 128 | [32, 64, 128, 256, 512] |
| Sampling interval | 1000 | [500, 1000, 2000] |
| Training time (per epoch) | 0.2s | |
| Inference time | $0.3s \sim 0.5s$ | |
| Trainable parameters | $0.75MB \sim 1.25MB$ | |

## D.3 DATA NORMALIZATION

Since there is no predefined end to a stream, meaning that reliable statistics (mean, variance, and maximum) do not exist, the data normalization in our implementation is operated on each individ-

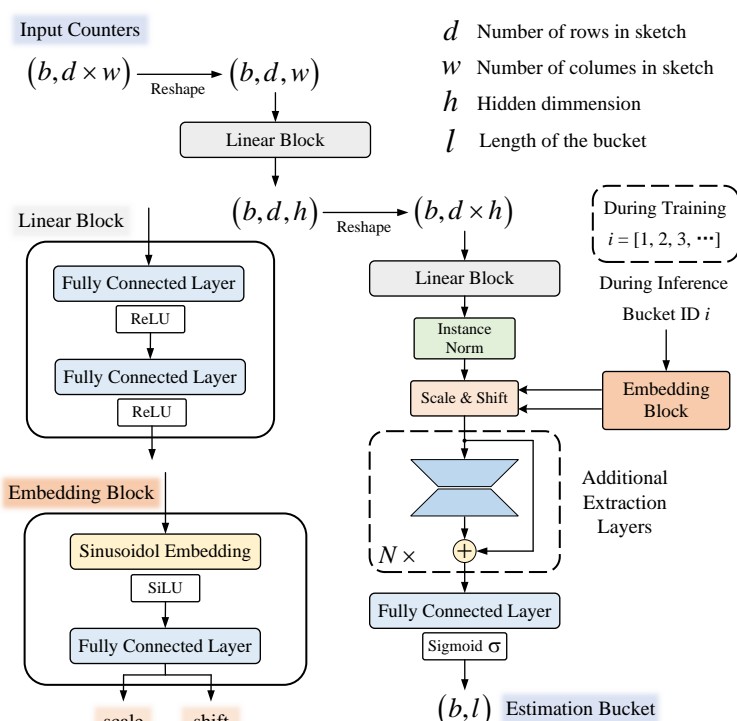

Figure 12: Neural network architecture of the learned solver used in our proposed approach.

ual sample in a batch separately. Now recalling the sketch update procedure, when a streaming item arrives, its volume is added to one counter in each row, where the counter is determined by $h_j$, $1 \leq j \leq d$. Therefore, counters in our sketch have the following guarantee: any inserted frequency should $\leq$ scale $:= \min_{1 \leq j \leq d} \max_{1 \leq i \leq w}$ sketch_count $[j, i]$. All we need is to find the minimum of the maximum counts from all the rows, which can be done in linear time. Then we calculate the instance-normalized measurement $y' = \frac{y}{\text{scale}}$, and the inverse transformation for final estimations can be written as $x = \text{scale} \times x_\theta$, in which $x_\theta$ is the output of the learned solver.

## D.4 BASELINES

As mentioned in Section 5.1, we compare our method with eight state-of-the-art sketch-based algorithms[2]: CM-sketch, C-sketch, Elastic Sketch, UnivMon, Nitrosketch, SeqSketch, Learned CM-sketch, and Learned C-sketch. For these original sketches, we manually configure it to approach the best memory-accuracy trade-off and allocate the same amount of memory for a fair comparison. To be more specific, the number of hash functions has been fixed at 4 across all methods. We fix levels as 2 in the universal sketch of UnivMon. For Elastic Sketch, we allocate $25\% \sim 50\%$ memory for its hash table. In particular, the neural network oracle in two learned sketches is replaced with an ideal oracle that knows the identities of the heavy hitters, whose target domain depends on the number of its unique buckets which take up around $50\%$ memory. We implement a version of Aamand et al. (2024)'s learned sketch which uses a single CS table. If the median estimate of an element is below a threshold of $CK/w$ for a tunable constant $C = 1$, the estimate is instead set to 0.

## D.5 MODEL DETAILS

The details of our neural network (NN) architecture are illustrated in Fig. 12. To recover an estimation bucket of length $l$, the size of the NN input measurements is $d \times w$. The measurement vector is first transformed to the shape $(d, w)$. After projection by a linear block, each containing two fully

---

[2]The implementation of all sketches (in Python) can be found in our codebase, which is written mainly based on a C++ Github repository: https://github.com/N2-Sys/BitSense.

connected layers (FCs) using ReLU Nonlinear, the size of the tensor returns to $d \times h$. Then, it will be fed into the second linear block. We adopt this design with the following motivations: (a) Considering the need to reduce the computational cost, the total number of parameters is several times less than that of its counterpart without transformations. (b) Motivated by the studies in sketching algorithms, hash (row) independence should have a certain regularization effect on the estimation process. Next, we integrate an embedding block to learn the bucket offset with information in the sketch. For the bucket ID, we employ Sinusoidal embedding to represent each $i$ as a $h$-dimensional vector, and then apply one fully connected layer after activating by SiLU Nonlinear. $i$ is injected into the network with $\text{scale}_i x + \text{shift}_i$, where $x$ is the shallow representation of our sampled counters. The fused feature is then extracted by a series of layers, e.g. FCs. Finally, features are projected back to the ground-truth domain with the Sigmoid function, as the normalized frequency is guaranteed to be $\leq 1$.

# E    OMITTED RESULTS OF FREQUENCY ENTROPY

In Table 4, we present experimental results of estimated entropy comparison on all datasets. Additionally, Table 5 provides a comparison of UCL-sketch and its variates with the same setting as before. As for Os and Ms, their entropy relative errors are both 126.49 on CAIDA, 489.29 on Kosarak, 618.81 on Retail, respectively.

Table 4: Entropy relative error on different streaming data sets (**bold** indicates best performance)

| Datasets | Sketches | 16KB | 32KB | 48KB | 64KB | 80KB | 96KB | 112KB | 128KB |
|---|---|---|---|---|---|---|---|---|---|
| CAIDA | Ours | **16.44** | **9.68** | **10.75** | **7.54** | **6.72** | **4.51** | **4.06** | **3.39** |
| | CM | 4117.71 | 1550.48 | 843.59 | 543.85 | 379.17 | 284.8 | 221.26 | 176.97 |
| | CS | 2781.76 | 1252.46 | 770.21 | 550.84 | 481.88 | 385.78 | 316.6 | 236.29 |
| | LCM | 7476.93 | 2475.28 | 937.02 | 806.84 | 442.46 | 394.74 | 239.47 | 227.62 |
| | LCS | 3042.2 | 1042.73 | 452.74 | 380.76 | 390.41 | 341.08 | 214.80 | 124.86 |
| | ES | 9745.67 | 3969.5 | 1486.76 | 1459.49 | 794.73 | 774.28 | 497.61 | 490.86 |
| | UM | 1820.53 | 1196.32 | 879.91 | 571.56 | 493.42 | 429.74 | 375.90 | 557.75 |
| | NS | 1187.6 | 510.67 | 305.52 | 212.99 | 162.92 | 131.79 | 107.70 | 105.86 |
| | LS | 6780.68 | 1001.38 | 502.98 | 276.07 | 32.22 | 21.54 | 16.60 | 13.42 |
| Kosarak | Ours | **58.24** | **35.10** | **21.63** | **16.85** | **14.29** | **13.19** | **10.79** | **10.18** |
| | CM | 2620.64 | 841.5 | 404.67 | 231.51 | 144.01 | 98.37 | 70.56 | 52.63 |
| | CS | 3270.70 | 1438.2 | 876.32 | 608.17 | 542.87 | 430.44 | 347.66 | 247.36 |
| | LCM | 4506.12 | 1008.47 | 295.55 | 196.1 | 86.35 | 60.62 | 27.47 | 23.44 |
| | LCS | 3469.50 | 846.16 | 317.96 | 202.46 | 173.8 | 119.25 | 61.44 | 34.50 |
| | ES | 6537.66 | 2268.03 | 703.97 | 643.72 | 299.46 | 274.27 | 155.77 | 144.44 |
| | UM | 1875.06 | 1222.37 | 902.99 | 565.21 | 524.43 | 483.68 | 393.32 | 654.84 |
| | NS | 3575.26 | 1532.43 | 914.97 | 625.38 | 462.48 | 361.92 | 294.88 | 288.81 |
| | LS | 7840.86 | 1887.36 | 683.64 | 439.85 | 227.36 | 154.53 | 79.38 | 34.03 |
| Retail | Ours | **69.16** | **41.02** | **27.14** | **22.20** | **18.87** | **15.07** | **12.56** | **10.56** |
| | CM | 3166.01 | 1066.05 | 511.37 | 291.21 | 179.20 | 119.71 | 82.79 | 59.45 |
| | CS | 3468.11 | 1569.19 | 984.25 | 690.76 | 643.02 | 516 | 419.62 | 305.66 |
| | LCM | 5527.20 | 1265.28 | 346.69 | 211.31 | 83.16 | 50.66 | 17.72 | 13.67 |
| | LCS | 4129.77 | 1097.03 | 415.05 | 254.86 | 208.78 | 130.63 | 60.03 | 32.82 |
| | ES | 7601.40 | 2621.57 | 850.06 | 742.85 | 344.02 | 282.78 | 156.78 | 129.85 |
| | UM | 1766.46 | 1197.95 | 894.09 | 582.79 | 591.83 | 494.88 | 448.04 | 773.58 |
| | NS | 3857.67 | 1734.13 | 1051.95 | 740.24 | 554.35 | 441.95 | 362.26 | 358.09 |
| | LS | 9349.86 | 2466.50 | 893.28 | 549.42 | 276.40 | 170.72 | 79.21 | 62.11 |

Table 5: Entropy relative error with viariates of UCL-sketch (**bold** indicates best performance)

| Viariates | UCL-sketch | w/o SA | w/o EQ | w/o SR | OMP | LSQR | CM |
|---|---|---|---|---|---|---|---|
| Relative Error | **16.85** | 408.93 | 376.53 | 358.09 | 346.78 | 301.32 | 204.29 |

## F   ADDITIONAL EXPERIMENTS

We supplement more extra experiments including ablational study, sensitivity studies and evaluation on synthetic datasets that are not presented in Section 5 due to the limit of space.

### F.1   ABALTIONAL STUDY

In Fig. 13 and Table 5 (see Appendix E), we compare the original UCL-sketch with its modified version to analyze the impact of each designed component. Our used dataset is Kosarak and memory is 64KB.

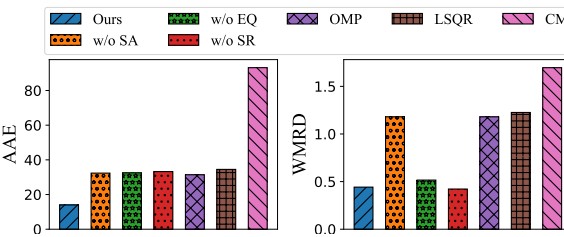

Figure 13: Quantitative results of ablational studies.

**Learning version v.s. Non-learning version.** Three traditional versions of the decoding method are used for comparison: OMP used in SeqSketch, LSQR (Paige & Saunders, 1982), and Count-Min (CM). As shown in Fig. 13, we can find that the accuracy under our learning-based algorithm is consistently better than that under non-learning methods. CM performs the worst since it does not utilize information from the linear system. Meanwhile, OMP is more precise than LSQR owing to its sparse greedy solution. We then measure its inference time for different numbers of keys. The left of Fig. 14 gives a recovery time of OMP that is over 60 seconds in some cases, which is much slower than ours which keeps the time around 0.5 seconds. Therefore, proposed learning technologies do boost the design of high-performance sketches.

**Basic training v.s. Optimized training.** Also see Fig. 13 for the ablation results on training options, whose details are as follows. *Without EQ*: We retrain the solver by removing the third term in Eqn. 2. Noticeable performance degradation is observed in both AAE and WMRD when compared to ours. This has indicated the effectiveness of equivalent learning for handling the null-space ambiguity without ground truth. *Without SR*: We remove the second loss term in Eqn. 2 and retrain the model. The accuracy is close to the situation without EQ. But it's interesting to see that WMRD after discarding the sparse regularization is slightly lower than the original version. The reason may be the sparsity assumption limits the model's ability to learn the heavy-tailed distribution weakly. However, Table 5 shows that UCL-sketch offers a much better estimation of frequency entropy than the other two variates for all memory sizes.

**Scalable network v.s. Non-scalable network.** To demonstrate the impact of *scalable architecture (SA)*, we train a network with unshared buckets. Surprisingly, the unshared version does not achieve the best performance in Fig. 13. This is because the parameter sharing acts as a form of regularization, preventing overfitting so that the solver is more likely

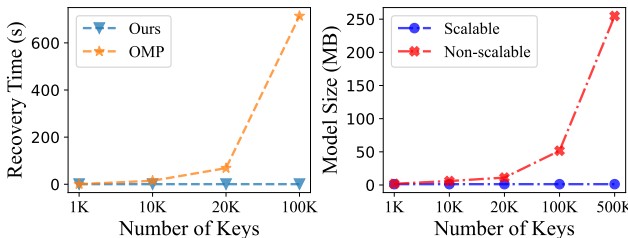

Figure 14: **Left:** Recovery time. **Right:** Size of parameters.

to generalize well to future counters, especially for very large-scale streams. A summary of the required memory is reported on the right of Fig. 14. Obviously, in the scalable network, with the increase in the number of keys, the parameters significantly decrease. By setting 500K unique items, the non-scalable network takes up a memory of over 250MB, while ours only requires a consumption of 1MB. This indicates that our compression scheme in parameters is very promising and practical, with an even better effect on estimation performance.

## F.2 SENSITIVITY ANALYSIS

We measure the influence of some key parameter settings, i.e. size of the bucket and hidden dimension, on accuracy, distribution, and resource usage. Also, we use the Kosarak dataset in these experiments.

### F.2.1 THE IMPACT OF BUCKET LENGTH

In this experiment, we vary bucket length in [128, 256, 512, 1024, 2056]. The results in Fig. 16 show that a small size of bucket can achieve similar AREs, but it leads to notable degradation in the entropy of estimated frequencies. Therefore, we choose 512 or 1024 as its value in other experiments to make a balance between overall performance and memory usage (see Fig. 15). In fact, users can set bucket length according to the key space they are interested in, and a length of 1024 will be enough in general.

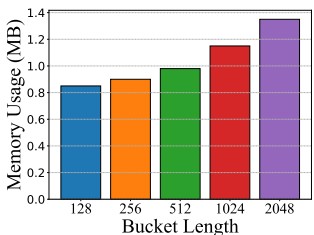

Figure 15: Total trainable parameters using 64KB sketch.

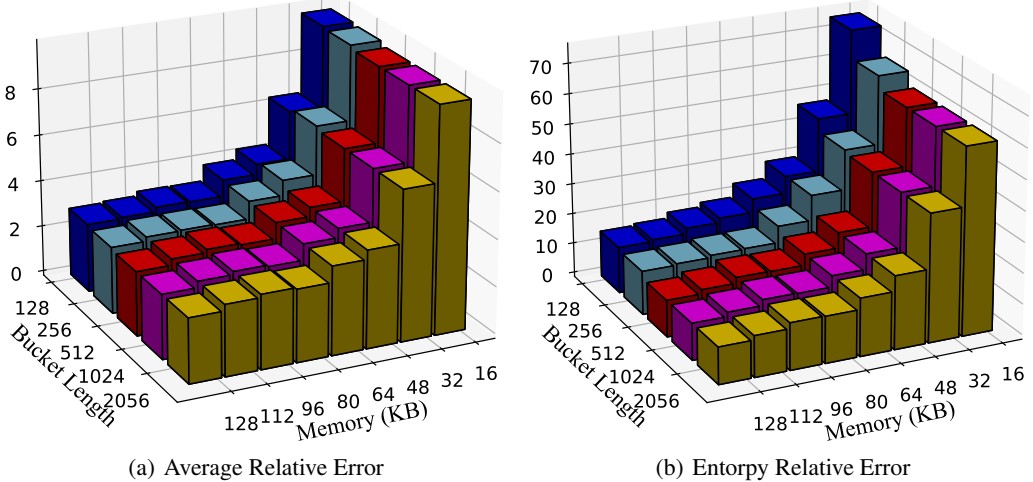

(a) Average Relative Error

(b) Entorpy Relative Error

Figure 16: Effects of the shared bucket length.

### F.2.2 THE IMPACT OF HIDDEN DIMENSION

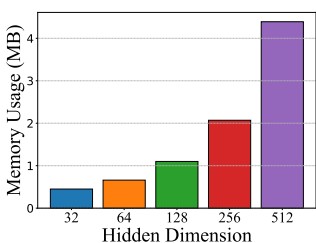

Figure 17: Total trainable parameters using 64KB sketch.

We retrain models with hidden dimensions from 32 to 512 to obtain 5×2 groups of experimental results in Fig. 18. As shown in Fig. 18, the performance of frequency estimation improves with increasing the hidden dimension, indicating that the higher-dimensional representation is available, the better the model can be trained and the better results can be achieved by the extracted feature. However, after the size reaches a certain value e.g. 128, the increase of dimension does not boost the recovery performance obviously. Also note that the memory overhead of the model exhibits exponential growth in high-dimensional (over 64) hidden spaces, which can be observed in Fig. 17. We thus set it to 128 in all experiments.

## F.3 ADDITIONAL RESULTS OF WEIGHTED ESTIMATION ERROR

Next, we also consider the following notion of overall weighted error as done in (Hsu et al., 2019) and (Aamand et al., 2024), which equals $\frac{1}{n}\sum_{i=1}^{n} f(i) \cdot |\hat{f}(i) - f^*(i)|$, where $f(\cdot)$ and $f^*(\cdot)$ are real and estimated frequency respectively. The weighted error is more natural from the machine learning perspective, as it can be interpreted as measuring the error with respect to the real frequency distribution. Fig. 19 plots the weighted errors of nine algorithms except for 0s and Ms, because they are generally larger than 300000. We see that three ideally learned sketches, e.g., LCM, LCS and

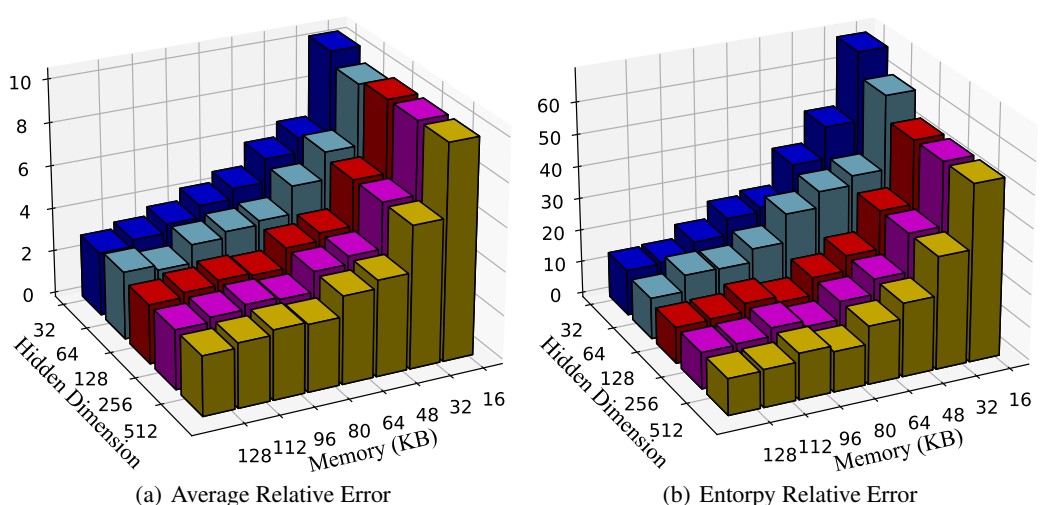

(a) Average Relative Error        (b) Entorpy Relative Error

Figure 18: Effects of the hidden dimension in the learned solver.

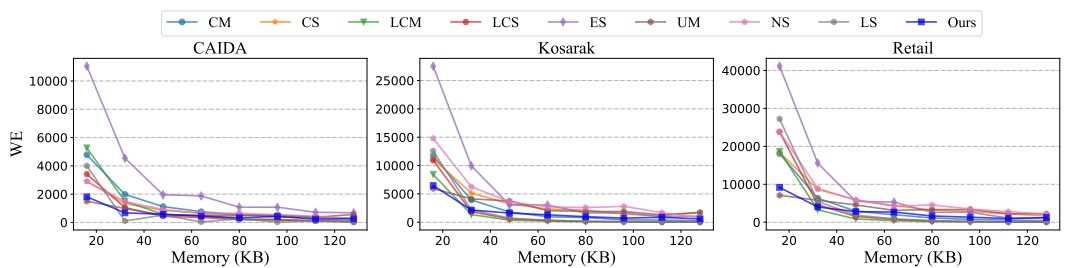

Figure 19: Comparison of weighted error on there real-world datasets.

LS, achieve obviously better estimation since they perfectly preserve the frequency of the largest error-weighted terms, and LS performs better in predicting the low-frequency components (set to 0). The practical performance of UCL-sketch is slightly inferior to theirs, but it remains highly competitive overall, achieving approximately an average 10% improvement compared to the best non-learning sketch.

## F.4 AREs OF DIFFERENT SKETCHES WITH MORE MEMORY USAGE

We now report additional details about the ARE experiments we show in the main body of the paper. In this section, we focus on the sketch performance with more memory usage. In Fig. 20, it becomes evident that when employing different sketches with a memory capacity exceeding 1MB, its ARE exhibit significant reductions, approaching approximately zero error. In particular, the figure shows

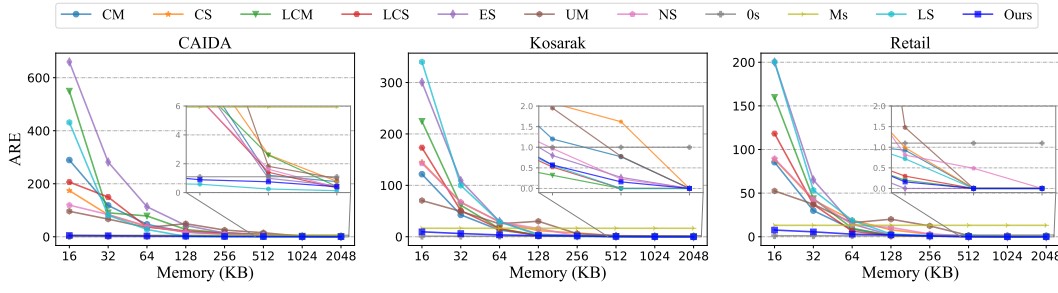

Figure 20: Comparison of AREs on three real-world datasets.

that learned sketch, i.e., LS achieves smallest error using memory over 512KB, while our GT-free approaches obtained results comparable to the supervised learning one.

## F.5 FREQUENCY ESTIMATION FOR TOY ZIPFIAN DATASETS

To evaluate the robustness of the proposed algorithm, we also synthesize four datasets that satisfy Zipf's law, where the skewness varies in [1.2, 1.3, 1.4, 1.5] and keys with length of 4-byte distinguish items in these datasets. There are 2M elements in each dataset, with around 22K ∼ 214K total distinct items depending on the skewness. Fig. 21 and Fig. 22 plot the average results regarding WE, AAE, ARE, and WMRD, while frequency entropy relative errors are listed in Table 6. The results confirm the conclusion shown in the main body of the paper: We observe that the performance of most methods increases as skewness rises.

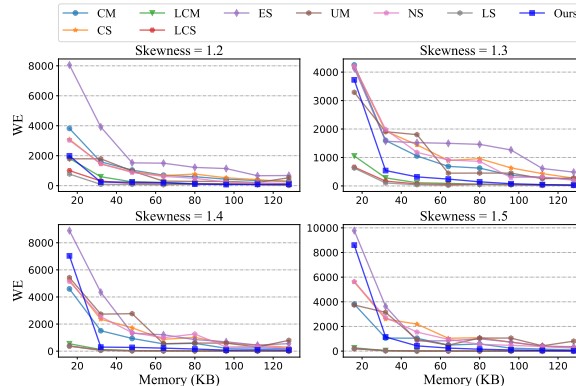

Figure 21: Comparison of weighted error on different Zipfian datasets.

This is because the heavy-tailed distribution leads to fewer keys given the same total count. Nevertheless, our UCL-sketch always performs the best in all memory settings. Note that even with the lowest skewness, the proposed algorithm can still achieve minimal deviation, e.g. average entropy relative errors never exceeding 9 in that case, beating the other algorithms with a significant advantage.

Table 6: Entropy relative error on different Zipfian datasets (**bold** indicates best performance)

| Skewness | Sketches | 16KB | 32KB | 48KB | 64KB | 80KB | 96KB | 112KB | 128KB |
|---|---|---|---|---|---|---|---|---|---|
| 1.2 | Ours | **8.18** | **6.23** | **5.53** | **2.87** | **2.58** | **3.56** | **4.64** | **3.01** |
| | CM | 3556.66 | 1354.36 | 757.90 | 502.69 | 359.71 | 277.20 | 220.82 | 180.74 |
| | CS | 4687.94 | 1869.68 | 1065.96 | 724.66 | 552.55 | 431.75 | 345.01 | 242.73 |
| | LCM | 6543.01 | 2290.03 | 921.88 | 830.31 | 481.52 | 448.20 | 295.59 | 288.44 |
| | LCS | 3107.83 | 1008.36 | 438.68 | 369.63 | 544.17 | 497.78 | 331.26 | 133.06 |
| | ES | 8435.34 | 3369.08 | 1286.89 | 1259.31 | 701.99 | 675.75 | 447.01 | 443.37 |
| | UM | 2362.29 | 1405.47 | 959.04 | 556.77 | 596.86 | 537.01 | 437.41 | 694.09 |
| | NS | 2816.48 | 1109.91 | 637.40 | 433.10 | 317.79 | 250.64 | 204.9 | 263.25 |
| | LS | 2854.41 | 490.34 | 418.59 | 237.34 | 270.75 | 217.02 | 105.18 | 50.12 |
| 1.3 | Ours | **3.70** | **3.26** | **2.64** | **1.41** | **1.54** | **2.29** | **3.19** | **1.60** |
| | CM | 1694.94 | 592.36 | 314.16 | 201.90 | 140.67 | 105.46 | 82.66 | 66.54 |
| | CS | 3189.71 | 1192.73 | 669.42 | 435.7 | 329.11 | 248.73 | 200.2 | 141.37 |
| | LCM | 2977.00 | 913.99 | 342.29 | 294.59 | 162.95 | 147.04 | 91.61 | 88.00 |
| | LCS | 1675.31 | 476.62 | 200.51 | 159.95 | 217.34 | 192.65 | 123.57 | 52.50 |
| | ES | 4258.16 | 1569.04 | 543.37 | 509.99 | 272.29 | 263.63 | 168.56 | 164.57 |
| | UM | 1858.68 | 1047.64 | 726.18 | 391.56 | 373.41 | 373.41 | 261.99 | 416.50 |
| | NS | 2160.49 | 816.45 | 454.08 | 296.26 | 210.27 | 167.69 | 132.00 | 167.98 |
| | LS | 1950.57 | 325.30 | 118.29 | 196.67 | 84.10 | 161.87 | 94.01 | 46.21 |
| 1.4 | Ours | **34.26** | **1.71** | **3.65** | **0.53** | **0.30** | **0.89** | **1.26** | **0.61** |
| | CM | 777.78 | 249.71 | 126.36 | 77.86 | 52.84 | 38.47 | 29.64 | 23.46 |
| | CS | 2015.95 | 724.19 | 394.16 | 289.24 | 188.41 | 137.07 | 115.87 | 83.55 |
| | LCM | 1303.13 | 338.55 | 116.05 | 93.28 | 48.16 | 41.25 | 23.69 | 22.19 |
| | LCS | 834.16 | 205.26 | 83.62 | 62.56 | 76.59 | 64.79 | 39.95 | 18.21 |
| | ES | 2106.11 | 660.07 | 213.24 | 200.60 | 100.88 | 94.11 | 58.06 | 55.11 |
| | UM | 1437.18 | 772.68 | 517.9 | 265.29 | 216.73 | 216.73 | 195.21 | 244.32 |
| | NS | 1528.59 | 525.88 | 278.60 | 174.61 | 131.69 | 96.45 | 81.31 | 102.76 |
| | LS | 982.51 | 156.91 | 56.34 | 63.64 | 23.80 | 46.63 | 23.91 | 20.09 |
| 1.5 | Ours | **24.69** | **2.94** | **4.86** | **4.25** | **1.71** | **4.02** | **4.46** | **0.79** |
| | CM | 357.16 | 105.53 | 50.26 | 29.70 | 19.52 | 13.76 | 10.51 | 7.98 |
| | CS | 1335.87 | 460.32 | 264.98 | 149.75 | 107.31 | 78.85 | 62.8 | 46.58 |
| | LCM | 554.13 | 117.42 | 36.07 | 26.13 | 12.23 | 9.47 | 4.73 | 4.22 |
| | LCS | 398.21 | 83.63 | 32.43 | 22.49 | 24.34 | 19.31 | 11.19 | 5.88 |
| | ES | 966.71 | 286.85 | 83.11 | 73.99 | 34.8 | 29.52 | 17.22 | 15.01 |
| | UM | 991.13 | 467.95 | 309.66 | 162.53 | 129.13 | 129.13 | 89.86 | 159.09 |
| | NS | 993.50 | 339.90 | 179.28 | 103.97 | 75.11 | 56.80 | 46.64 | 58.71 |
| | LS | 464.66 | 56.41 | 20.48 | 6.77 | 16.61 | 9.74 | 5.29 | 4.07 |

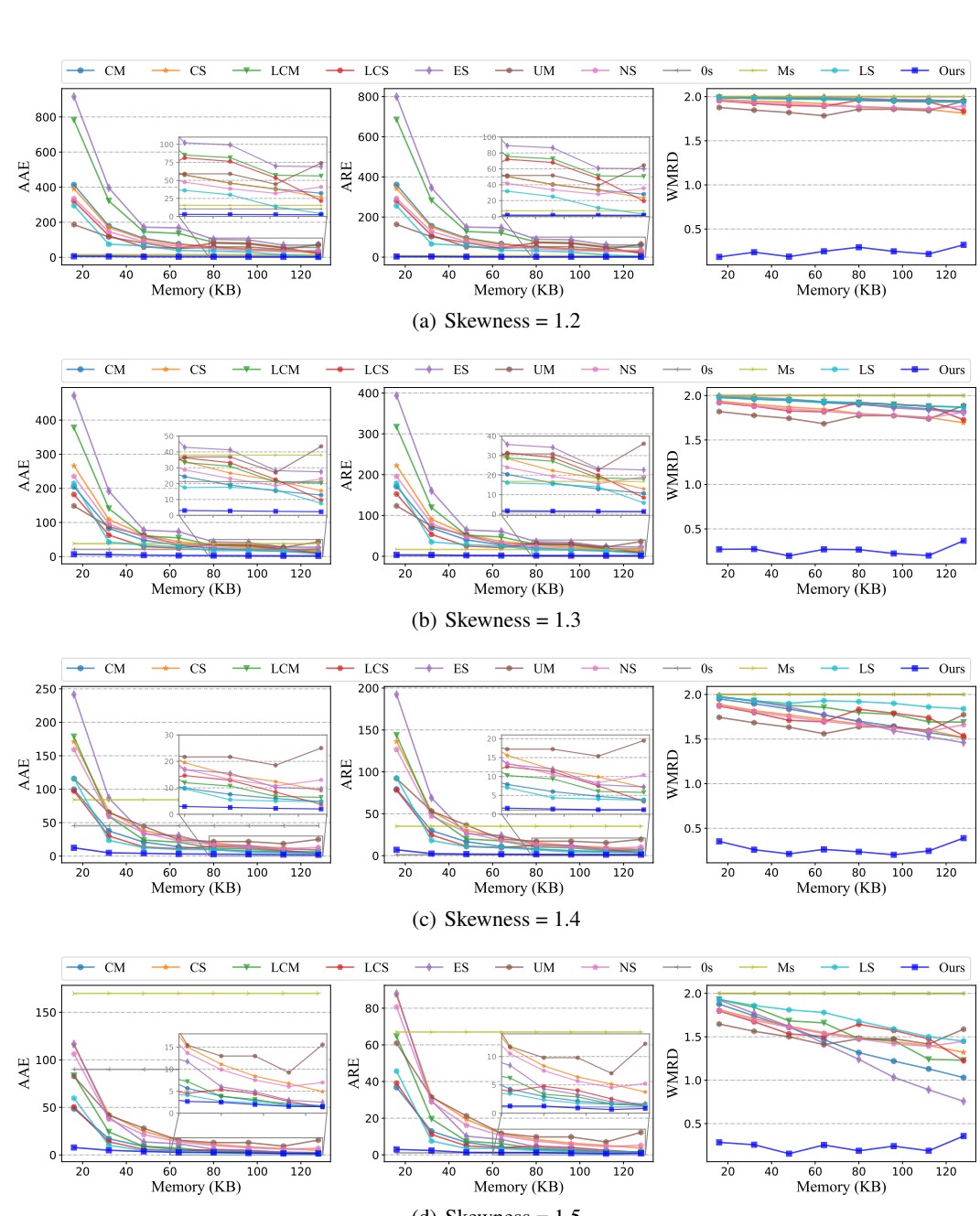

Figure 22: Performance comparison between our UCL-sketch and existing state-of-the-art sketches on synthetic Zipfian datasets with different skewness.

