# OpenReview forum: "Toward Practical Learning-based Frequency Estimation without Ground Truth"
_ICLR.cc/2025/Conference — ICLR 2025 Conference Withdrawn Submission_

### Official Review · Reviewer_hzBt · 2024-11-02

**Soundness:** 2
**Presentation:** 3
**Contribution:** 2
**Rating:** 6
**Confidence:** 3

**Summary:**

The paper studies the problem of learning-augmented frequency estimation on a stream. The goal is to maintain a sketch that can approximately report the frequency of the elements seen on the stream so far. A more recent line of work improves the accuracy/memory trade-off by augmenting the sketch with an ML component, trained on past frequency data.

The sketch proposed in this paper trains the ML component online as the stream is observed (as opposed to prior approaches that assumed access to an already trained ML component at the start of the stream). In order to avoid having to rely on unknown ground truth frequencies, the ML component is trained on approximate frequencies obtained by compressed sensing.

**Strengths:**

Training on the fly and without access to ground truth is indeed more realistic and useful than training on ground truth data prior to the beginning of the stream. The experimental results show that the proposed method performs favorably compared to baselines.

**Weaknesses:**

The analysis seems to yield weaker results than in prior works. In Hsu et al, Aamand et al, for example, the error guarantee applies directly to the implemented algorithm. Here, in theorem 2, the guarantee is given under the idealized assumption that the ML component learns a "perfect" solution $f^*$. This disregards the specific learning dynamics (particularly in an online setting) and the fact that such ideal $f^*$ might not be representable by the ML component at all. This doesn't render the analysis wrong or meaningless, albeit weaker in nature and less directly related to the method than in prior works on the problem.

**Questions:**

If the premise of theorem 2 is that $Af=Af^*$ (you use exact equality in some steps of the proof), why do you have $\|Al\|\approx0$ (where $l=f^*-f$) and not just $\|Al\|=0$ in other parts, and what does "$\approx$" precisely mean here anyway?

---

> ### Author Response · Authors · 2024-11-19
> **Response to Reviewer hzBt**
>
> We thank the reviewer for taking the time reviewing our work. Our responses are below.
>
> > The analysis seems to yield weaker results than in prior works. In Hsu et al, Aamand et al, for example, the error guarantee applies directly to the implemented algorithm. Here, in theorem 2, the guarantee is given under the idealized assumption that the ML component learns a "perfect" solution $f^*$. This disregards the specific learning dynamics (particularly in an online setting) and the fact that such ideal $f^*$ might not be representable by the ML component at all. This doesn't render the analysis wrong or meaningless, albeit weaker in nature and less directly related to the method than in prior works on the problem.
>
> **A4.1:** For results on previous learning-augmented sketches, they initially assume that the heavy hitter oracle is perfect, i.e., that it makes no mistakes when classifying the heavy items or correctly identifying the top $\mathcal{O}(K)$ heavy elements. Although the specific on-line learning dynamics is not considered in Theorem 2, the offline approach of Hsu et al and Aamand et al. also does not account for potential distribution shifts, such as frequent items becoming infrequent in the future. Thus, Therefore, we do not agree that our analysis is weaker than previous work. In fact, as evidenced by our strong empirical results, our algorithm effectively generalizes to noisy real-world predictions, demonstrating the robustness of the learning component used in our work.
>
> > If the premise of theorem 2 is that $Af=Af^*$ (you use exact equality in some steps of the proof), why do you have $|Al| \approx 0$ (where $l=f-f^*$) and not just $|Al| = 0$ in other parts, and what does "$\approx$" precisely mean here anyway ?
>
> **A4.2:** Thank you for the question! Our "assumption" or key ponit here is: After being sufficiently trained with the guidance of the designed objective loss including the term $|Af-Af^*|$, $|Al|=|Af-Af^*|$ can be very small which $\approx 0$ for scenarios without noise in measument verctor $y=Af$. We believe this is a natural and very mild premise, and similar trick can be found in [1] (cited for sure).
>
> **References:**
> > [1] Candes, Emmanuel J. "The restricted isometry property and its implications for compressed sensing." Comptes rendus. Mathematique 346.9-10 (2008): 589-592.

---

> > ### Comment · Reviewer_hzBt · 2024-11-22
> >
> > Thank you for the reply.
> >
> > > For results on previous learning-augmented sketches, they initially assume that the heavy hitter oracle is perfect
> >
> > It's the difference between proving that the algorithm as is works in a certain (arguably idealized) setting, versus proving that an idealized variant of the algorithm (rather than the algorithm itself) works. But fair enough, there isn't a direct comparison here.
> >
> > > "assumption" or key ponit here is:
> >
> > I understand you assume that the learned solution $f^*$ is as good as the ground truth $f$ (this was the subject of the previous question). My question was what are you formally assuming here in the proof, and I'm still not sure. Are you just assuming that $|Af-Af^*|=0$ (in which case the "$\approx0$" notation seems redundant and confusing, and I would consider dispensing with it), or is there is a specific formal meaning to "$\approx0$". I am also not sure what part in [1] you mean, a more specific pointer inside it would help if relevant.

---

> > > ### Author Response · Authors · 2024-11-22
> > >
> > > > My question was what are you formally assuming here in the proof, and I'm still not sure. Are you just assuming that $|Af-Af^*|=0$ (in which case the "$\approx0$" notation seems redundant and confusing, and I would consider dispensing with it), or is there is a specific formal meaning to "$\approx0$". I am also not sure what part in [1] you mean, a more specific pointer inside it would help if relevant." and agree to dispense "$\approx0$".
> > >
> > > Thank reviewer for raising these follow-up points. You are correct that the $\approx 0$ notation might be redundant and potentially confusing. To clarify, in the proof, we are assuming $f^*$ minimize $\left\| {{f^*}} \right\|_1$ subject to $\boldsymbol{A}f^*=\boldsymbol{A}f$. Given this, I agree that the $\approx 0$ notation can be dispensed with, as it doesn’t add clarity in this context. We have updated the proof to reflect this and remove the unnecessary notation by direct setting $\left\| \boldsymbol{A}f^* - \boldsymbol{A}f \right\|_2=0$. Regarding [1], I appreciate your feedback. The specific part I was referring to is [Lemma 2.2, page 4, "Indeed, this follows from ... with $\epsilon = 0$."]. Thanks again for your thoughtful input, and let me know if there’s anything else that needs clarification!

---

### Official Review · Reviewer_wtry · 2024-11-02

**Soundness:** 2
**Presentation:** 1
**Contribution:** 2
**Rating:** 3
**Confidence:** 4

**Summary:**

This paper proposed a new frequency estimation streaming algorithm within the framework of learning-augmented algorithms. Past work on learning-augmented frequency estimation (e.g., Hsu et al, 2019) uses a machine learned predictor of heavy hitters trained on past data to improve the performance of classic sketches such as CountMin and CountSketch. The approach in this paper is quite different. The starting point is equation-based sketches for frequency estimation which recover an estimate of the frequency vector of the stream by attempting to invert the sketching process (this is an underdetermined problem but one can hope to find sparse solutions). The authors posit that existing methods are too cumbersome and propose learning a function to produce a frequency estimate from the sketch. They propose to learn this method online as the stream arrives and choose a loss function based on a heavy-tailed assumption of the frequencies. They provide experimental evidence that their proposed sketch performs well in terms of error and computation compared to baselines.

**Strengths:**

The experimental performance of the proposed sketch is impressive. It is shown to have high throughput and achieve low error. Recovery time is good relative to other equation-based solutions without learning (though recovery using non-equation-based sketches is much simpler and likely much faster).

The idea of using learning to speed up the inversion process of equation-based skeches is interesting and fits nicely in the learning-augmented algorithms framework.

**Weaknesses:**

The paper is difficult to read. The goals of the algorithm are not made clear from the introduction and problems/solutions of the design process are introduced alongside each other in an ad hoc manner in reading the paper. There are many moving parts described in the paper, but the reader is not adequately guided through them. The algorithmic choices are generally not clearly motivated and  explanations for why the proposed solutions are the right ones are missing. I would suggest simplifying the paper into clear sections which each address a specific and well-motivated question about the design of the algorithm. For instance, an interesting question is how to incorporate learning to speed up the inversion procedure of equation-based sketches. Clearly and formally stating this problem and its solution could be a self-contained section. Another question is how to do this learning efficiently over the course of a stream. Separately defining these challenges, possible baselines, and your solution could constitute a second focus on the paper. Currently, all these components are muddled together in the text.

The experiments are missing a comparison to the improved learning augmented sketching algorithms presented in Aamand et al. (2023) That work demonstrates that simple tricks (such as truncating low-frequency esimates to zero) yields significant improvements over LCS. Both that paper and the original by Hsu et al. (2019) measure error in terms of expected error weighted by the frequency of keys, which is missing from the experimental results. This corresponds to a query distribution matching the distribution of the stream. Furthermore, in the Aamand et al. paper, they discuss parsimonious learning-augmented sketches which get the same guarantees while infrequently querying the machine learning oracle. This would have a significant affect on throughput as it is currently measured in this paper by running inference on an RNN at every timestep.

**Questions:**

- What are the sizes in bytes of the datasets?

 - What fraction of space is used for the different components of your algorithm for a fixed space budget?

 - Does the Bloom filter component of the algorithm limit the algorithm to have a minimum space requirement growing linearly with the number of unique keys?

 - Why is the throughput of CM and CS less than UCL? If appropriately implemented, I would suspect that CM and CS have very fast throughput. Is the training component of UCL counted in the throughput?

---

> ### Author Response · Authors · 2024-11-19
> **Response to Reviewer wtry (Part I)**
>
> We thank the reviewer `wtry` for the careful reading of the paper and extensive comments. We have addressed the questions below.
>
> > The paper is difficult to read. The goals of the algorithm are not made clear from the introduction and problems/solutions of the design process are introduced alongside each other in an ad hoc manner in reading the paper. There are many moving parts described in the paper, but the reader is not adequately guided through them. The algorithmic choices are generally not clearly motivated and explanations for why the proposed solutions are the right ones are missing. I would suggest simplifying the paper into clear sections which each address a specific and well-motivated question about the design of the algorithm. For instance, an interesting question is how to incorporate learning to speed up the inversion procedure of equation-based sketches. Clearly and formally stating this problem and its solution could be a self-contained section. Another question is how to do this learning efficiently over the course of a stream. Separately defining these challenges, possible baselines, and your solution could constitute a second focus on the paper. Currently, all these components are muddled together in the text.
>
> **A3.1:** Thank reviewer for the valuable feedback. We appreciate your insights on the clarity and structure of the paper. As you suggested, we have improved the organization to enhance the readability of our paper (marked in blue). To be more specific, we removed redundant content and set a separate discussion in methodology part to explain our detailed motivations. Moreover, at the begining of each design of the algorithm, we have added a description of the problem to be solved. Finally, we put the basic design and our optimizations together to build the complete version of UCL-sketch. Thank you again for the insightful comments — We believe these changes strengthen the paper’s clarity and impact.
>
> > The experiments are missing a comparison to the improved learning augmented sketching algorithms presented in Aamand et al. (2023) That work demonstrates that simple tricks (such as truncating low-frequency esimates to zero) yields significant improvements over LCS.
>
> **A3.2:** In the revised manuscript, we added the comparison with the learning-augmented algorithms presented by Aamand et al. We provide detailed experimental results and analysis to illustrate how our approach performs relative to their methods. This comparison is discussed in experimental section and additional results are available in the appendix. We also provide estimation results when all predictions are set to 0 or to the mean frequency.
>
> > Both that paper and the original by Hsu et al. (2019) measure error in terms of expected error weighted by the frequency of keys, which is missing from the experimental results. This corresponds to a query distribution matching the distribution of the stream.
>
> **A3.3:** Thank you for pointing that out. We have added the relevant results, and experiments on the weighted mean error can be found in the Appendix F.3.
>
> > Furthermore, in the Aamand et al. paper, they discuss parsimonious learning-augmented sketches which get the same guarantees while infrequently querying the machine learning oracle. This would have a significant affect on throughput as it is currently measured in this paper by running inference on an RNN at every timestep.
>
> **A3.4:** In the Aamand et al. paper, they only query the heavy-hitter oracle with some probability $p$. Following their setting: queries occur $\mathcal{O}(K)$ times, we repeat throughput experiments by randomly running the neural network $1 \times K \sim 10 \times K$ times, instead of querying the entire stream. Below, we present a table comparing the average throughput of our method with theirs.
>
> |Method|4bite|8bite|13bite|
> |:--------:|:--------:| :---------:|:--------:|
> |Ours|76682 $\pm$ 1960|40587 $\pm$ 1551|23100 $\pm$ 1198|
> |Aamand et al.|16080 $\pm$ 2325|11507 $\pm$ 1889|5400 $\pm$ 974|
>
> > What are the sizes in bytes of the datasets ?
>
> **A3.5:** Except for the CAIDA dataset which uses 13 bytes, all other datasets, including the generated Zipfian data, use 4 bytes.
>
> > What fraction of space is used for the different components of your algorithm for a fixed space budget ?
>
> **A3.6:** When the space is small, the proportion of the Bloom Filter is larger (over 50%), but there is a memory limit (for example, lower than 10KB for Kosarak and Retail dataset). As the space becomes more abundant, we gradually increase the allocation to the hash table, but HF: sketch overall 1:1 as shown in Table 2.

---

> ### Author Response · Authors · 2024-11-19
> **Response to Reviewer wtry (Part II)**
>
> > Does the Bloom filter component of the algorithm limit the algorithm to have a minimum space requirement growing linearly with the number of unique keys ?
>
> **A3.7:** Yes, the Bloom filter component of the algorithm can theoretically introduce a minimum space requirement that grows with the number of unique keys, but the growth is not strictly linear in practical. For our equation-based sketch, a false positive does not introduce significant estimation errors; instead, it simply leads to the neglect of some most infrequent keys, without negatively impacting the system's overall performance (as you mentioned, truncating low-frequency esimates to zero still yields good performance). This is because we store the hottest and second-hottest (or evicted) keys in the hash table and hot-key set, respectively. Our experiments confirm this, showing that a low memory overhead does not substantially affect accuracy. Additionally, similar data structures incorporating Bloom Filters have been deployed successfully in real-world streaming scenarios, achieving both high accuracy and efficient resource usage as demonstrated in prior studies [1,2,3].
>
> > Why is the throughput of CM and CS less than UCL ? If appropriately implemented, I would suspect that CM and CS have very fast throughput. Is the training component of UCL counted in the throughput ?
>
> **A3.8:** From A-sketch [4]: A non-learning heavy filter "utilizes the skew of the underlying stream data to improve the frequency estimation accuracy for the most frequent items by filtering them out earlier. It also improves the overall throughput — while using exactly the same amount of space as traditional sketches.". That means given the heavy-tailed distribution of streaming data, the heavy filter (or hash table) absorbs a large portion of insertion operations. Therefore, sketch equipped with filters is efficient since updating cost in the hash table is cheap. In fact, Elastic Sketch achieves the best throughput, mainly because it has more insertion operations taking place within its multi-level filters than ours.
>
> **References:**
> > [1] Huang, Qun, et al. "Toward Nearly-Zero-Error sketching via compressive sensing." 18th USENIX Symposium on Networked Systems Design and Implementation (NSDI 21). 2021.
> [2] Sheng, Siyuan, et al. "PR-Sketch: monitoring per-key aggregation of streaming data with nearly full accuracy." Proceedings of the VLDB Endowment 14.10 (2021): 1783-1796.
> [3] He, Jintao, et al. "Histsketch: A compact data structure for accurate per-key distribution monitoring." 2023 IEEE 39th International Conference on Data Engineering (ICDE). IEEE, 2023.
> [4] Roy, Pratanu, et al. "Augmented sketch: Faster and more accurate stream processing." Proceedings of the 2016 International Conference on Management of Data. 2016.

---

> > ### Comment · Reviewer_wtry · 2024-11-19
> > **Preliminary response by reviewer**
> >
> > Thanks for the detailed responses to the points raised in my and the other reviews. It will take me some time to read the new additions to the paper, but here are some quick questions.
> >
> > - I was interested actually in the total size in bytes of the datasets, not the size of a single key. The x-axes of the plots are in KB and I want a sense for what fraction of the space required to store the whole stream is used in those plots. Could you please provide this info?
> > - For the throughput calculations for the learned sketches, I just want to make sure that for the elements currently predicted to be heavy, the learned predictor is not run on these elements, correct? In fact, all that should happen for these elements is that they are found in a lookup table (could even use a Bloom filter if throughput is what you care about) and their counts are incremented by one. In that sense, the parsimonious learned algorithm should have an opportunity of having improved throughput over the basic CountSketch depending on the querying probability and the sketch size for the filtering reason you mention.
> > - My interpretation is that $K$ is the number of distinct elements in the stream. If this is the case, then it is not true that the number of queries of the parsimonious learned algorithm needs to grow linearly with $K$, rather, the total number of queries should grow linearly with the space of the streaming algorithm (in particular, the number of predicted heavy elements you wish to store).

---

> ### Author Response · Authors · 2024-11-20
> **Response to Reviewer wtry (Part III)**
>
> Dear reviewer `wtry`, thank you for taking the time to read our responses. Regarding your quick question,
>
> > I was interested actually in the total size in bytes of the datasets, not the size of a single key. The x-axes of the plots are in KB and I want a sense for what fraction of the space required to store the whole stream is used in those plots. Could you please provide this info?
>
> **A3.9:** Thank reviewer for the clarification. We have provided the relevant information in the dataset description section: the lengths of the data streams for the three real-world datasets are approximately 1 million each, and the total length of every generated dataset is around 2 million. So considering for the CAIDA dataset which uses 13 bytes, all other datasets use 4 bytes, the total size in bytes of the datasets is as follows: CAIDA:12695KB, Kosarak:3906KB, Retail:3554KB and Zipfian data: 7812KB.
>
> > For the throughput calculations for the learned sketches, I just want to make sure that for the elements currently predicted to be heavy, the learned predictor is not run on these elements, correct? In fact, all that should happen for these elements is that they are found in a lookup table (could even use a Bloom filter if throughput is what you care about) and their counts are incremented by one. In that sense, the parsimonious learned algorithm should have an opportunity of having improved throughput over the basic CountSketch depending on the querying probability and the sketch size for the filtering reason you mention.
>
> **A3.10:** We think there may be a misunderstanding regarding the definition of the throughput metric presented in the paper. We agree the parsimonious learned algorithm should have an improved throughput over CountSketch during querying stage. But notice that, from Section 5.2 PERFORMANCE COMPARISON: "We perform **insertions** of all items in a stream, record the total time used, and calculate the throughput." which means that *we focus on the speed of record insertion rather than query processing* because, in scenarios where the data stream rate is extremely high (such as network traffic monitoring or log analysis), insertion speed is often critical. For example, in network monitoring scenarios, if the insertion speed cannot keep up with the incoming data rate, it may result in data loss or delays.

---

> > ### Author Response · Authors · 2024-11-20
> >
> > Now I understand: there is also a filter that absorbs most of it, but when we directly ideally removed the top-k large flows (perfect prediction), and only performed insertion operations for the small flows using two-layer RNN, the result is still not very fast.

---

> ### Author Response · Authors · 2024-11-20
> **Response to Reviewer wtry (Part IV)**
>
> > My interpretation is that $K$ is the number of distinct elements in the stream. If this is the case, then it is not true that the number of queries of the parsimonious learned algorithm needs to grow linearly with $K$, rather, the total number of queries should grow linearly with the space of the streaming algorithm (in particular, the number of predicted heavy elements you wish to store).
>
> **A3.11:** Thank you for reminding me of the definition of $K$, we will correct this throughput error ! (Now it's fixed, we set $p$=5% here.)

---

### Official Review · Reviewer_1ws2 · 2024-11-03

**Soundness:** 2
**Presentation:** 3
**Contribution:** 2
**Rating:** 5
**Confidence:** 4

**Summary:**

This paper presents an unsupervised learning-based frequency estimation algorithm UCL-sketch which is ground truth free, thus it can be learned in online scenarios and quickly response to streaming distribution drift. And UCL-sketch realizes highly scalable architecture with logical buckets sharing parameters across them. Experiments on real datasets demonstrate the UCL-sketch having better accuracy compared to baselines.

**Strengths:**

1. The paper is well written and organization is clear
2. Unsupervised online learning of estimating the frequency of items seems interesting and practical.
3. The experimental results highlight some benefits of the proposed solution.

**Weaknesses:**

1. As an unsupervised learned sketch, this paper seems to overlook some important related works, such as ICML19 [1], ICLR23[2], and TPAMI24 [3].  These studies also utilize unsupervised learning to construct learned sketches for summarizing data streams, which appears to be quite similar to the approach in this paper. It would be better if technical comparison on these works can be made to highlight the specific contributions of this work.

2. Although this paper demonstrates the effectiveness of UCL-sketch on certain error metrics, the error values appear relatively high. For example, in terms of ARE, several sketches cited in this paper generally achieve values below 1 with suitable memory settings. This expectation aligns with the definition of ARE, where an overall ARE of 1 would result if all predictions were zero. Consequently, an ARE exceeding 1 may lack practical significance. The authors should consider adjusting the memory settings to bring the error into a more reasonable range for evaluation.

[1]Rae et al. "Meta-learning neural bloom filters. " International Conference on Machine Learning. PMLR, 2019.

[2]Feng, et al. "Mayfly: a Neural Data Structure for Graph Stream Summarization." The Twelfth International Conference on Learning Representations. 2023.

[3]Cao, et al. "Learning to Sketch: A Neural Approach to Item Frequency Estimation in Streaming Data." IEEE Transactions on Pattern Analysis and Machine Intelligence (2024).

**Questions:**

Please address weakness points 1-2.

What are the specific values of AAE, ARE, and WMRD errors under different data streams when all predictions are set to 0 or to the mean frequency?

Others:
There is a typo in the statements of contributions: 'UCL-sketch' is mistakenly written as 'ULC-sketch.'

---

> ### Author Response · Authors · 2024-11-19
> **Response to Reviewer 1ws2**
>
> We thank the reviewer `1ws2` for the valuable feedback; our responses are below. We are happy to continue the discussion in case more questions arise.
>
> > As an unsupervised learned sketch, this paper seems to overlook some important related works, such as ICML19 [1], ICLR23[2], and TPAMI24 [3]. These studies also utilize unsupervised learning to construct learned sketches for summarizing data streams, which appears to be quite similar to the approach in this paper. It would be better if technical comparison on these works can be made to highlight the specific contributions of this work.
>
> **A2.1:** We thank reviewer for the references which are indeed interesting and belong to the specialized field of learning-enhanced data structure. We have added the references to the literature review and discuss them accordingly in Appendix/Related Work. Regarding the comparison you mentioned, we have looked into them further. However, We believe that [1] (Bloom Filter) and ICLR23 [2] (data structure for graph streams) focus on areas that are not entirely aligned with the scope of this work; As for the latest work [3] published in TPAMI recently, their code does not appear to be available. So implementing these methods with adequate benchmarking is not feasible during the discussion period.
>
> > Although this paper demonstrates the effectiveness of UCL-sketch on certain error metrics, the error values appear relatively high. For example, in terms of ARE, several sketches cited in this paper generally achieve values below 1 with suitable memory settings. This expectation aligns with the definition of ARE, where an overall ARE of 1 would result if all predictions were zero. Consequently, an ARE exceeding 1 may lack practical significance. The authors should consider adjusting the memory settings to bring the error into a more reasonable range for evaluation.
>
> **A2.2:** We appreciate your suggestion to explore alternative memory settings and have incorporated additional experiments on AREs in the appendix to evaluate UCL-sketch under configurations (over 1MB, details see Fig. 20) that achieve more competitive ARE values, ensuring a more comprehensive comparison.
>
> > What are the specific values of AAE, ARE, and WMRD errors under different data streams when all predictions are set to 0 or to the mean frequency?
>
> **A2.3:** We agree setting all predictions to 0 or the mean frequency represents baseline strategies that can provide a reference for evaluating sketch performance. Therefore, we added the comparison in experimental section and additional results are available in the appendix.
>
> > Others: There is a typo in the statements of contributions: 'UCL-sketch' is mistakenly written as 'ULC-sketch.'
>
> **A2.4:** Thank the reviewer for pointing out these minor writing mistakes. We carefully revised our manuscript and modified the typos in the revised manuscript.
>
> **References:**
> > [1] Rae, Jack, Sergey Bartunov, and Timothy Lillicrap. "Meta-learning neural bloom filters." International Conference on Machine Learning. PMLR, 2019.
> [2] Feng, Yuan, et al. "Mayfly: a Neural Data Structure for Graph Stream Summarization." The Twelfth International Conference on Learning Representations. 2023.
> [3] Cao, Yukun, et al. "Learning to Sketch: A Neural Approach to Item Frequency Estimation in Streaming Data." IEEE Transactions on Pattern Analysis and Machine Intelligence (2024).

---

> > ### Comment · Reviewer_1ws2 · 2024-11-21
> >
> > Thanks for your reply. I still have concerns about the presented evaluation. In Figure 8, there is no observable WMRD error corresponding to predictions that are entirely zeros. Is this an unintentional oversight, or does it indicate that the error at this point lies outside the range depicted on the y-axis?
> >
> > Furthermore, as shown in Figure 8 of the main experiment, under most memory configurations, nearly all algorithms fail to surpass the performance of a simple predictor that outputs all zeros. This observation underscores the inadequacy of using such an evaluation as the primary experiment, as it provides little practical insight for real-world applications.
> >
> > In sketch-related papers cited in this paper, there are a few alternative approaches to address such issues:
> >
> > 1. Exclude memory budgets that are too restrictive, ensuring the primary budget range maintains the error within a reasonable threshold (e.g., ARE < 1) [1].
> >
> > 2. Employ a logarithmic scale for the y-axis to represent the error range more effectively [2].
> >
> > So, I recommend revising the main experiment settings to ensure that all sketches are evaluated under practical memory configurations. Such adjustments are essential for achieving meaningful and fair comparisons.
> >
> > References:
> >
> > [1] Tong Yang, Jie Jiang, Peng Liu, Qun Huang, Junzhi Gong, Yang Zhou, Rui Miao, Xiaoming Li, and Steve Uhlig. Elastic Sketch: Adaptive and Fast Network-Wide Measurements. Proceedings of the 2018 ACM SIGCOMM Conference, pp. 561–575, 2018.
> >
> > [2] Pratanu Roy, Arijit Khan, and Gustavo Alonso. Augmented sketch: Faster and more accurate stream processing. In Proceedings of the 2016 International Conference on Management of Data, pp.1449–1463, 2016.

---

> > > ### Author Response · Authors · 2024-11-21
> > >
> > > Dear Reviewer 1ws2,
> > >
> > > We sincerely appreciate your reviews and valuable suggestions. Regarding the WMRD for all-zero predictions, we apologize for selecting an inconspicuous marker, which led to your misunderstanding. Its real value completely overlaps with the estimates using all-means, i.e., the maximum value of 2.0. In our revision, we have replaced the marker to address this issue. Moreover, apart from ARE, other metrics (AAE, WMRD, Entropy, and Weighted Error) show that the performance of all-zero predictions is not particularly strong (under almost all memory settings, it is outperformed by our method). As shown in Fig.20, we conducted experiments under broader memory usage conditions. When the space exceeds 512KB, almost all methods surpass the all-zero predictor.
> > >
> > > Best regards,
> > >
> > > The Authors.

---

> > > > ### Comment · Reviewer_1ws2 · 2024-12-03
> > > >
> > > > Thank you for your feedback. I would like to keep the score as it is.

---

### Official Review · Reviewer_FPNi · 2024-11-04

**Soundness:** 2
**Presentation:** 3
**Contribution:** 2
**Rating:** 3
**Confidence:** 5

**Summary:**

The paper studies learning-augmented algorithms for frequency estimation in the streaming model. As far as I can tell (see more below), the main idea seems to be training a heavy-hitter prediction as the streaming algorithm progresses. This is an interesting idea and departs from the methodology used in prior works, which assume a heavy-hitter prediction oracle that has been already trained on past data (and whose space does not need to be accounted for as it will be `ammortized' across many data streams).

**Strengths:**

See below.

**Weaknesses:**

Unfortunately I believe there are some fundamental misunderstandings in the paper, which I hope to clarify below.

1) Understanding of prior works: It is claimed that all sketching based frequency estimation algorithms can all be captured by solving a linear system. This is simply incorrect, even for the two most famous examples, CountMin and CountSketch. While it is true that they are both linear sketches in the sense that their updates are just linear equations, the estimation part of the algorithms are not linear! Take CountMin for example. CountMin hashes the universe into a small number of buckets and repeats this independently (this part is all linear). But when it comes to the estimation phase (the authors call it recovery), it is crucial for the theoretical guarantees that one takes the *minimum* estimated frequency across all independent trials. This is not a linear operation. Similarly for CountSketch, we need to take the median across independent trials.

2) Space complexity: I am not sure if the theoretical guarantees of the proposed algorithm are meaningful. The space useage seems to be *linear* in the domain size. E.g. on line 349 K seems to be the size of the universe and the algorithm uses linear in $K$ space. Thus, theoreticaly, this offers no advantage over just keeping all the exact frequencies in memory.

2) Error metric: The authors do not seem to realize that prior learning-augmented works for frequency estimation are not using the standard point-wise error (they are not bounding the worst error across all frequency estimates). Rather, they are bounding a notion of weighted error (see Eq. 3.1 in Hsu et al https://openreview.net/pdf?id=r1lohoCqY7). This is also true for the follow up works, such as Aamand et al (NeurIPS 23). This does not seem to be the case here. That is totally fine, as the total sum of error used in this work is also natural. But, this means that directly comparing to prior work as done in Table 1 does not make sense. Furthermore, the error bounds derived for prior work, as in Aamand et al., are specialized to the Zipfian frequency case and are also normalized (so the frequencies add to 1). Neither of those are true for this paper so I am confused why the authors are directly comparing to such bounds in Table 1.

3) Empirical evaluations: The paper fails to compare against the state of the art learning-augmented algorithms of Aamand et al. (NeurIPS 23).

4) Writing: Finally, I don't really understand what the exact proposed algorithm is. I've tried to read the submission several times and while the high level idea is intriguing, there are no clear details given in the submission. A formal algorithm block in the main body with all parts self-contained would be helpful towards this. In the theoretical section, there are many parameters used with no explanations. For example, what are $\epsilon_c, \delta_c$ in line 346?

5) The authors do not seem to realize that countmin and countsketch also have guarantees based on the `tail' contribution of the frequency vectors. E.g. see Lemma 4.1.5 here (https://www.sketchingbigdata.org/fall20/lec/notes.pdf). This is similar to what Theorem 2 is claiming (I think), but using sublinear space (whereas the submission seems to use linear in the universe size space ...).

Given these points, while the high level idea seems promising, it is hard for me to judge the contribution of this paper and recommend acceptance.

**Questions:**

See above.

---

> ### Author Response · Authors · 2024-11-19
> **Response to Reviewer FPNi (Part I)**
>
> We are glad the reviewer `FPNi` found our proposal is an interesting idea and departs from the methodology used in prior works. We address the raised concerns below. We are delighted to continue the discussion in case of additional questions.
>
> > Understanding of prior works: It is claimed that all sketching based frequency estimation algorithms can all be captured by solving a linear system. This is simply incorrect, even for the two most famous examples, CountMin and CountSketch. While it is true that they are both linear sketches in the sense that their updates are just linear equations, the estimation part of the algorithms are not linear! ...
>
> **A1.1:** We really appreciate the reviewer for the feedback regarding the estimation framework and apologize for any formulations that may lead to misunderstandings. It is reasonable to point out that the estimation part of Count-Min and Count Sketch are nonlinear. However, here we focus on the goal of sketch-based estimation, and the differences among the mentioned sketches lie solely in the way they approximate this goal: Count-min Sketch uses a minimum values as results, Count Sketch uses median values as results, and our method involves linear reconstruction for estimate. In other words, the target of the all above sketches is to obtain the true value $\boldsymbol{A}^{+} y+(I-\boldsymbol{A}^{+} \boldsymbol{A})x$, which is the solution to the linear system constructed by its $\boldsymbol{A}$. Furthermore, as indicated by the title of Section 2, we here only disscuss equation-based sketches, which are a type of linear sketch (see page 2 in Graham Cormode https://people.cs.umass.edu/~mcgregor/711S12/sketches1.pdf). We have made corresponding modifications to Theorem 1. In fact, the compressive sensing properties of various sketches have already been discussed in previous works such as [1], and similar statement to ours can be found in [2].
>
> > Space complexity: I am not sure if the theoretical guarantees of the proposed algorithm are meaningful. The space useage seems to be linear in the domain size. E.g. on line 349 $K$ seems to be the size of the universe and the algorithm uses linear in space. Thus, theoreticaly, this offers no advantage over just keeping all the exact frequencies in memory.
>
> **A1.2:** Thank the reviewer for pointing out this issue. First, we think the time and space complexity are both meaningful factors in evaluating the efficiency and scalability of algorithms and data structures. Second, although the spatial complexity is proportional to the size of the key space, the actual space required can be significantly less. You are correct that the space complexity is theoretically linear in the domain size as represented by $K$, owing to the Bloom Filter because a very low false positive rate (see Lemma 2) requires a larger bit array and more hash functions, which increases memory usage. But considering its application in data sketching [3], Bloom Filters can offer great advantages over keeping all exact keys in memory, particularly when some false positives are acceptable. It makes senses in our UCL-sketch as the consequence of a false positive is not the introduction of a remarkable error in computation, but likely only ignores some most infrequent items that does not adversely impact the overall performance of the system (as shown in Aamand et al. (NeurIPS 2023), reporting low-frequency esimates as 0 yields even better result), because we actually performed two rounds of filtering: once through a hash table and once through an additional hot-key set. Our experiments demonstrate this: lower memory cost, such as 16KB, does not have a significant impact on the accuracy. Besides, the similar equation-based sketch (with a Bloom Filter) has been deployed in real-world streaming scenarios and proven to achieves both high accuracy and low resource usage in prior works [1,4,5].
>
> **References:**
> > [1] Huang, Qun, et al. "Toward Nearly-Zero-Error sketching via compressive sensing." 18th USENIX Symposium on Networked Systems Design and Implementation (NSDI 21). 2021.
> [2] Fu, Yongquan, et al. "Clustering-preserving network flow sketching." IEEE INFOCOM 2020-IEEE Conference on Computer Communications. IEEE, 2020.
> [3] Cormode, Graham. "Data sketching." Communications of the ACM 60.9 (2017): 48-55.
> [4] Sheng, Siyuan, et al. "PR-Sketch: monitoring per-key aggregation of streaming data with nearly full accuracy." Proceedings of the VLDB Endowment 14.10 (2021): 1783-1796.
> [5] He, Jintao, et al. "Histsketch: A compact data structure for accurate per-key distribution monitoring." 2023 IEEE 39th International Conference on Data Engineering (ICDE). IEEE, 2023.

---

> ### Author Response · Authors · 2024-11-19
> **Response to Reviewer FPNi (Part II)**
>
> > Writing (Part I): Finally, I don't really understand what the exact proposed algorithm is. I've tried to read the submission several times and while the high level idea is intriguing, there are no clear details given in the submission. A formal algorithm block in the main body with all parts self-contained would be helpful towards this.
>
> **A1.5.1:** We feel sorry for the lack of clarity regarding the proposed approach. Based on the reviewer’s suggestion, we added a summary section (see Section 3.4) and illustrated our overall algorithm through Fig.7 in the main body of the revised manuscript, ensuring that all components and steps are clearly defined and self-contained.
>
> > Writing (Part II): In the theoretical section, there are many parameters used with no explanations. For example, what are $\epsilon_c,\delta_c$ in line 346?
>
> **A1.5.2:** Thank you for your feedback. In response, we have rewritten the *notation* part of Section 4. Particularly, $(\epsilon_c,\delta_c)$ are standard error formulations of traditional sketches. For example, in original CM-sketch paper (see Theorem 1 of http://dimacs.rutgers.edu/~graham/pubs/papers/cm-full.pdf), they are coefficient and error probability such that estimation error $\le \epsilon_c\|f\|$ with probability at least 1 - $\delta_c$. Actually, $\epsilon_c,\delta_c$ are determined by the depth $d$ and width $w$ of the sketch counters, as defined in our paper.

---

> ### Author Response · Authors · 2024-11-19
> **Response to Reviewer FPNi (Part III)**
>
> > The authors do not seem to realize that countmin and countsketch also have guarantees based on the `tail' contribution of the frequency vectors. E.g. see Lemma 4.1.5 here (https://www.sketchingbigdata.org/fall20/lec/notes.pdf). This is similar to what Theorem 2 is claiming (I think), but using sublinear space (whereas the submission seems to use linear in the universe size space ...)..
>
> **A1.6:** First, we list the lemma mentioned by the reviewer: The CM-sketch with $4d \le w$ guarantees that for any $i$, $|f^*(i) - f(i)| = \|f_{T_0^c}\|_1/d$ with probability $\ge$ 1 - $\delta_c$. Then given $\|f^*-f\|_1=\sum\nolimits_i |f^*(i) - f(i)|$, the lemma indicates $\|f^*-f \| _1$ has a value of $\frac{K}{d} \|f _{T_0^c}\|_1$ with a small probability.
>
> Now, considering the underdetermined scenario: $w \times d = M < K$ with $w, d \ge 1$ in the paper, it is evident that our error upper bound is lower than that of the CM-sketch, i.e., $\|f _{T_0^c}\|_1$  $\le$ $w \|f _{T_0^c}\| _1$ $= \frac{M}{d} \|f _{T_0^c}\|_1$ $< \frac{K}{d} \|f _{T_0^c}\|_1$. Finally, we really appreciate your comment, which has allowed us to better situate our work within the current literature.

---

### Note · Authors · 2024-12-06

I have read and agree with the venue's withdrawal policy on behalf of myself and my co-authors.